# HERO: HARNESSING TEMPORAL MODELING FOR DIFFUSION-BASED VIDEO OUTPAINTING

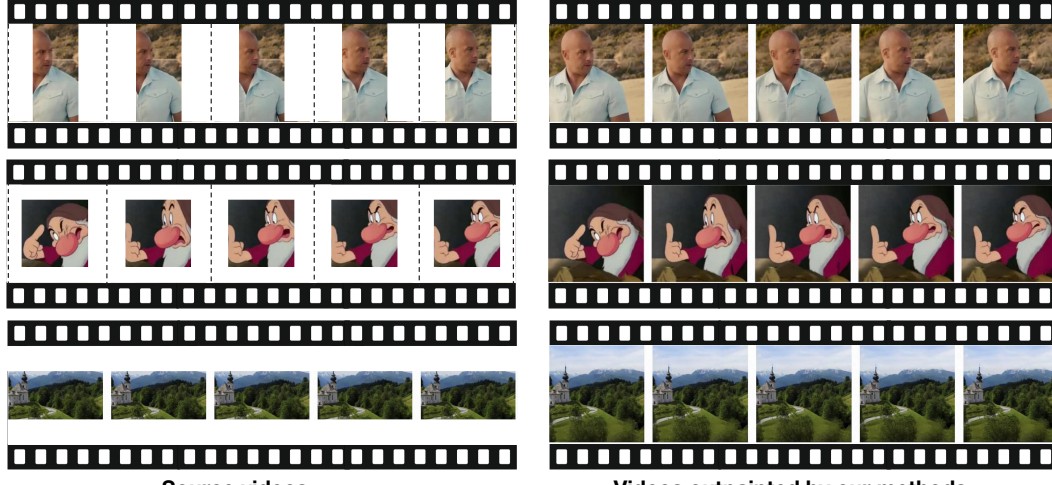

Source videos          Videos outpainted by our methods

Figure 1: Video outpainting results by HERO. They include Vertical Outpainting, Central Outpainting and Horizontal Outpainting results on character portraits, cartoons and landscape videos. More interesting videos can be found in the supplementary materials.

## ABSTRACT

Video outpainting expands the spatial perspective of a video, enabling it to adapt to various display devices with different aspect ratios. Current diffusion-based approaches for video outpainting often suffer from quality issues such as blurred details, local distortion, and temporal instability, significantly impacting the user experience. The root cause is the insufficient temporal modeling in video outpainting, which inadequately represents the relationships between frames over time. To address this issue, a novel approach called HERO (**H**arnessing the t**E**mpo**R**al modeling for diffusion-based **O**utpainting) is proposed to effectively tackles these generated video quality problems. HERO employs two critical components to enhance temporal modeling: the Temporal Reference Module, which provides reference features that extend beyond spatial dimensions; and the Interpolation-based Motion Modelling Module, designed to stabilize generated frames. By integrating these modules, these quality issues in video outpainting are effectively addressed. Extensive experiments on multiple benchmarks demonstrate that HERO outperforms existing methods qualitatively and quantitatively.

## 1 INTRODUCTION

Video outpainting (Yu et al., 2023; Fan et al., 2023; Wang et al., 2024a) expands a video's spatial scope beyond its original perspective, enabling it to adapt to various screen ratios for diverse display devices and occasions. Unlike image outpainting, which focuses on a single frame, video outpainting must ensure both content consistency and spatial-temporal coherence to avoid jitter between adjacent frames. Differing from video inpainting, which focuses on internal areas with rich context and has a small mask ratio, video outpainting often deals with larger mask areas at frame edges with limited

(a) Input Videos for video outpainting.

(b) Videos expanded by M3DDM and MOTIA.

(c) Distribution of common reference features.

(d) The vanilla motion modeling module.

Figure 2: The video quality issues and their origins in current diffusion-based methods. (a) Videos to be expanded. (b) Expanded videos by current diffusion-based methods show blurred details and local distortion. (c) The commonly used reference features visualized by t-SNE (Van der Maaten & Hinton, 2008). They do not occupy the entire feature plane. (d) The vanilla motion modeling module performs global attention across all frames without considering the adjacent relations among frames.

context. For instance, the mask ratio on the DAVIS dataset (Caelles et al., 2019) is less than 20% for inpainting tasks (Zhou et al., 2023), while it reaches 66% for outpainting tasks (Fan et al., 2023). These complexities pose a greater challenge for video outpainting, thereby attracting considerable research interest recently.

Research on video outpainting can be categorized into mask-based and diffusion-based approaches. The mask-based methods (Yu et al., 2023) outpaint videos by predicting the masked content using a BERT-like (Devlin et al., 2018) learning approach. However, these methods rely only on contextual video tokens and are unable to generate high-definition videos. On the other hand, diffusion-based methods leverage significant advancements in image (Ho et al., 2020a; Rombach et al., 2022; Zhang et al., 2023) and video (Guo et al., 2024; Wang et al., 2023; Hu et al., 2023) generation achieved by diffusion models. These methods frame video outpainting as a video-to-video generation task (Fan et al., 2023; Wang et al., 2024a). These approaches have achieved state-of-the-art (SoTA) results.

Despite their advancements, SoTA diffusion-based video outpainting approaches (Fan et al., 2023; Wang et al., 2024a) still face several problems. These include blurred details, local distortion as shown in Fig. 2b, and temporal instability, which are evident in the interesting videos provided in the supplementary materials. A manual analysis of the output videos generated by M3DDM (Fan et al., 2023) on datasets DAVIS (Caelles et al., 2019) and YouTube-VOS (Xu et al., 2018) revealed that 38% of the videos are blurry, as detailed in Sec. C. These issues significantly reduce the audience's experience and impact the information delivery. The primary cause behind these problems is **insufficient temporal modeling** in video outpainting. This refers to an inadequate representation of temporal relationships between frames, leading to frame distortion or instability. The SoTA methods (Fan et al., 2023; Wang et al., 2024a) currently use reference conditions such as VAE (Kingma & Welling, 2013) for textual features and CLIP (Radford et al., 2021) for semantic features. However, *these features are all spatial dimensions and do not include any temporal reference features*. Fig. 2c shows that VAE and CLIP features only occupy the red and pink regions, leaving the cyan and blue regions empty. The widely used VAE and CLIP references are paradigms for static image synthesis and editing, which are insufficient for video. Moreover, current video generation methods (Guo et al., 2024; Hu et al., 2023; Fan et al., 2023; Wang et al., 2024a) depend on a vanilla motion modeling module (Guo et al., 2024), which performs global attention at the feature pixel level across all frames. This approach overlooks the relationships between adjacent frames and leads to temporal instability in generated videos. Such oversight further underscores the inadequacies in temporal modeling.

The challenges in solving the above problems are twofold. Firstly, there are limited established methods available for incorporating temporal references in video generation. Consequently, researchers must explore innovative approaches to better capture effective temporal references. Secondly, any improvements to the motion modeling module in the diffusion network should be minimized to maximally preserve the internal knowledge of the pre-trained diffusion network.

To overcome these significant challenges, 3D features that capture video-level information and optical flow features that convey motion dynamics are taken into consideration to enhance temporal

modeling. Features from these two perspectives encapsulate information beyond spatial dimensions and compensate for the limitations of the CLIP and VAE models. Additionally, the interpolation technique can be integrated into the vanilla motion modeling module to enhance the stability of the generated frames with few learnable parameters. Based on these ideas, a pioneering approach effectively **H**arnessing the t**E**mpo**R**al modeling for diffusion-based **O**utpainting (HERO) is proposed to handle the generated video quality problems. In HERO, a Temporal Reference Module is introduced in addition to the spatial-based reference modules (VAE and CLIP), providing comprehensive reference features. Subsequently, an Interpolation-based Motion Modeling Module is designed with **a single learnable scalar** to *enhance the stability of generated videos*.

The key contributions of this paper can be summarized as follows: (1) To the best of our knowledge, this is the first paper to comprehensively address the insufficient temporal modeling problem in diffusion-based video outpainting methods. Simultaneously, this paper demonstrates its causes, great impact, and challenges in addressing them. (2) The proposed HERO can alleviate the insufficient temporal modelling problem through the Temporal Reference Module, which provides comprehensive temporal references, and the Interpolation-based Motion Modeling Module, which stabilizes the generated frames. (3) HERO is validated through extensive quantitative and qualitative experiments, achieving state-of-the-art performance on multiple video outpainting benchmarks.

## 2 METHODOLOGY

### 2.1 PRELIMINARIES

**Stable Diffusion**. Our approach extends Stable Diffusion (SD) which is derived from the latent diffusion model (LDM) (Rombach et al., 2022). SD consists of a VAE (Kingma & Welling, 2013) and a UNet (Ronneberger et al., 2015) augmented by the cross-attention mechanism (Vaswani et al., 2017). VAE consists of an encoder $\mathcal{E}$ and a decoder $\mathcal{D}$. The encoder $\mathcal{E}$ of VAE first transforms an image from pixel space into a low-dimensional latent space to reduce the computational complexity for UNet: $\mathbf{z} = \mathcal{E}(\mathbf{x})$. During the training process of SD, the image latent $\mathbf{z}_0$ is diffused in $T$ time steps to produce noise latent $\mathbf{z}_T$. Simultaneously, a denoising UNet is trained to predict the applied noise. The optimization process is defined as follow function:

$$\mathcal{L}_{LDM} = \mathbb{E}_{\mathbf{z}_t, \epsilon \sim \mathcal{N}(0,1), t, c}[\|\epsilon - \epsilon_\theta(\mathbf{z}_t, c, t)\|_2^2], \tag{1}$$

where $\epsilon$ is the noise added to $\mathbf{z}_0$, $c$ denotes the conditional information and $t$ is the time step, $\epsilon_\theta$ represents the denoising UNet. In each iteration, the denoising UNet predicts noise on the latent feature for each timestep $t$. During inference, $\mathbf{z}'_T$ is sampled from a random Gaussian distribution at timestep $T$ and progressively denoised to $\mathbf{z}'_0$ using a guided sampling process (*e.g.*, DDPM (Ho et al., 2020b), DDIM (Song et al., 2021)). Finally, decoder $\mathcal{D}$ reconstructs image $x' = \mathcal{D}(\mathbf{z}'_0)$.

**Video Outpainting.** Let $v \in \mathbb{R}^{t \times h \times w \times 3}$ denotes an input video, where $t$ is the number of frames in the video, $h$ and $w$ are the height and width of the video and 3 stands for the channel number. Video outpainting extends the initial height and width of a video to a specified height and width, resulting in a new video $v' \in \mathbb{R}^{t \times h' \times w' \times 3}(h' > h, w' > w)$. Video outpainting needs to maintain both content consistency and spatial-temporal coherence. In the video, the initial perspective is referred to as the *known region*, while the extended area is termed as the *unknown region*.

### 2.2 MODEL OVERVIEW

HERO contains the Temporal Reference Module and Spatial Reference Module. Fig. 3 (I) shows the Temporal Reference perspective. The input video is padded to meet the target height and width (from $h \times w$ to $h' \times w'$) and is then sent into VAE to obtain the 4-channel latent feature $f_l$. $f_l$ is sent to the 3D Reference Net (3D-RefNet) to obtain the **video-level features**. The padded video is also transformed into optical flow maps and then concatenated with binary masks. They are then sent to the Optical Flow Encoder for **motion features**. Fig. 3 (II) shows the Spatial Reference perspective. The latent feature $f_l$ is concatenated with noise and binary masks. The padded video is also sent to the CLIP for semantic features. These reference features are sent repsectively to the 3**D-UNet** which is commonly used in video generation works (Guo et al., 2024; Hu et al., 2023; Fan et al., 2023; Wang et al., 2024a). Fig. 3 ⓐ shows the Interpolation-based Motion Modeling Module conducted on the temporal dimension within adjacent frames with learnable weights.

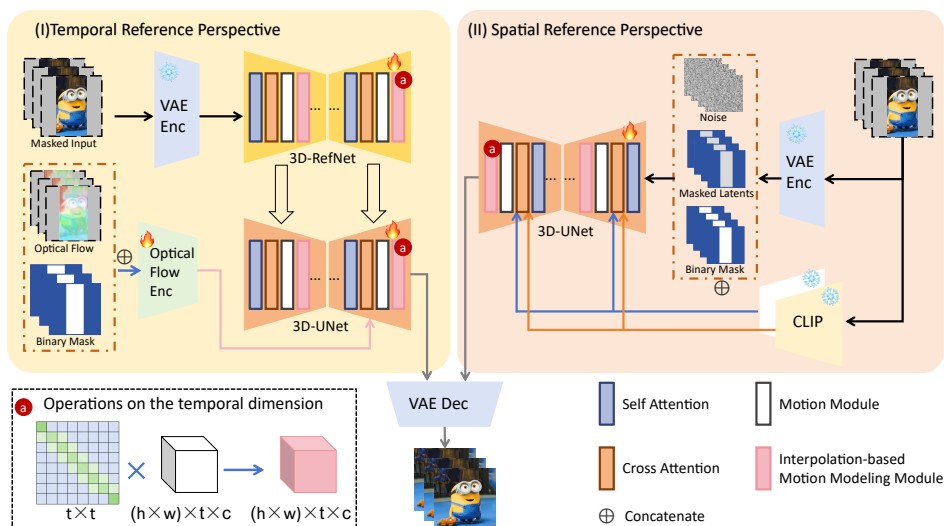

Figure 3: The overview of HERO. (I) The temporal reference perspective of HERO. (II) The spatial reference perspective of HERO. **Two perspectives share the same** 3**D-UNet**. ⓐ is the Interpolation-based Motion Modeling Module conducted on the temporal dimension for frames stability.

## 2.3 TEMPORAL REFERENCE MODULE

### 2.3.1 3D-REFNET

3D-RefNet is designed to extract **video-level features**, inspired by the ReferenceNet (Hu et al., 2023) which was originally designed for image reference. The structure of 3D-RefNet is almost identical to that of the 3D-UNet, except that the input is 4-channel for 3D-RefNet while for 3D-UNet it is 9-channel. The input to 3D-RefNet is the latent features extracted by the VAE model, without any other information concatenated. The weights of 3D-RefNet come from Stable Diffusion (Rombach et al., 2022) and the weights of the motion modeling module come from AnimateDiff (Guo et al., 2024). During the forward phase, the feature map $v_1 \in \mathbb{R}^{t \times h \times w \times c}$ from 3D-RefNet and the feature map $v_2 \in \mathbb{R}^{t \times h \times w \times c}$ from 3D-UNet are concatenated along $w$ dimension. Then a self-attention is performed on this concatenated feature map and the first half of the feature map serves as the output as in (Hu et al., 2023). During the training phase, the weights of 3D-RefNet and 3D-UNet are updated independently of each other. It should be noted that 3**D-RefNet and ReferenceNet for image reference differ in the following three respects**: in terms of network structure, 3D-RefNet adds a motion modeling module to capture video-level information; in terms of input, the input 3D-RefNet is multiple frames rather than a single reference image; and in terms of internal operations, 3D-RefNet does not require the tiling and copying of feature maps to align feature dimensions.

### 2.3.2 OPTICAL FLOW ENCODER (OFE)

Optical flow is widely used in video completion tasks (Zhou et al., 2023; Dehan et al., 2022). It reflects pixel-level **motion features**, which provides an alternative perspective for temporal information compared with 3D-RefNet. This kind of information can be beneficial for video generation and thus needs a dedicated encoder. On the other hand, ControlNet controls the generation of images with a condition image in a fine-grained manner. The main network structure of ControlNet is identical to the encoder in 3D-UNet, which is connected to 3D-UNet through the zero-initialized convolutional layer. The popular ControlNets support inputs include edge maps, pose key points, and segmentation maps, *etc.*, but they do not support the optical flow. Therefore, a ControlNet is trained from scratch to serve as the encoder for optical flow. Specifically, the dense optical flow of the input video is first estimated and the unknown regions will be filled with zero. It is then concatenated with a binary mask indicating the known and unknown regions along with channel dimensions to form the input.

### 2.4 INTERPOLATION-BASED MOTION MODELING MODULE (IM³)

The vanilla motion modeling module is proposed in (Guo et al., 2024) and widely used in video generation works (Fan et al., 2023; Wang et al., 2024a; 2023; Hu et al., 2023; Tian et al., 2024). The input feature map of vanilla motion modeling module is reshaped from $x \in \mathbf{R}^{b \times c \times f \times h \times w}$ into



(a) $\mathbf{M}_1$        (b) $\mathbf{M}_2$        (c) $\mathbf{M}_3$        (d) $\mathbf{M}_I$

Figure 4: In Fig. 4a, 4b, and 4c, the green regions are assigned a value of 1, while the values of all other regions are set to 0. Fig. 4d is a composite of 4a, 4b, and 4c according to Eq. 2.

$x \in \mathbf{R}^{(b \times h \times w) \times f \times c}$, where $b$ denotes the batch size, $h$ and $w$ are the height and width of the feature map, $f$ stands for the frame number and $c$ is the feature dimension. The vanilla motion modeling module then performs a temporal global self-attention across frames. However, *the global attention making senses in spatial dimension to capture long-range dependencies, may not make sense in temporal dimension.* It is common sense that the closer the two frames are, the more similar they become. The global attention without such prior introduces more noise, resulting in jitter between adjacent frames.

To handle this problem, the Interpolation-based Motion Modeling Module is proposed which is implemented with a learnable aggregation kernel shown in Fig. 4d. This kernel consists of three parts, *i.e.*, $\mathbf{M}_1, \mathbf{M}_2$ and $\mathbf{M}_3$ as shown in Fig. 4(a-c):

$$\mathbf{M}_I = \alpha \mathbf{M}_1 + (1 - 2\alpha)\mathbf{M}_2 + \mathbf{M}_3, \tag{2}$$

where $\mathbf{M}_1, \mathbf{M}_2, \mathbf{M}_3 \in \mathbb{R}^{t \times t}$ are all binary mask and $\alpha$ is a learnable scalar. When $\alpha \rightarrow 0$, the Interpolation-based Motion Modeling Module behaves more like an identity process. When $\alpha \rightarrow 1$, it exhibits the behaviour of an interpolation process.

And then, $\mathbf{M}_I$ multiplies the features $\mathbf{F} \in \mathbb{R}^{(b \times h \times w) \times t \times d}$ from the vanilla motion modeling module to obtain the refined feature map $\mathbf{F}' \in \mathbb{R}^{(b \times h \times w) \times t \times d}$ as shown in Eq. 3 and in Fig. 3 ⓐ.

$$\mathbf{F}' = \mathbf{M}_I \times \mathbf{F}. \tag{3}$$

Each of the other frames is enhanced by its preceding and following frames. This approach fully leverages the prior knowledge in the temporal domain. The learnable $\alpha$, has minimal modifications to the network structure to keep the latent space of Stable Diffusion, enabling the network to determine the optimal weights between itself and its neighbours.

## 2.5 SPATIAL REFERENCE MODULE

Spatial references are indispensable in video outpainting to complete individual frames. The spatial features are mainly drawn from the CLIP (Radford et al., 2021) and VAE (Kingma & Welling, 2013) as shown in Fig. 3 (II), as established in most diffusion-based methods (Fan et al., 2023; Wang et al., 2024a; Ye et al., 2023; Li et al., 2024; Wang et al., 2024b; Shi et al., 2023; Xiao et al., 2023).

**CLIP features.** The CLIP proposed by OpenAI consists of an image encoder and a text encoder, where the image encoder is a ViT (Dosovitskiy et al., 2020). The OpenAI CLIP weights are trained on a wide variety of image and text pairs which are abundantly available on the internet. Under the supervision of the text, the image feature from OpenAI CLIP contains the semantics of each frame and thus is adopted in this work. In addition, the Open CLIP (Ilharco et al., 2021) is an open source implementation of CLIP (Radford et al., 2021) trained on LAION-2B (Schuhmann et al., 2022). It achieves better performance on the ImageNet benchmark and is adopted in modern image generation methods, such as IP-Adapter (Ye et al., 2023). Inspired by the design of dual text encoders design in the SDXL (Podell et al., 2023), the features of these two image encoders are both kept and feed into the 3D-UNet using the decoupled cross-attention mechanism (Ye et al., 2023) shown as follows.

$$\mathbf{Z}' = \text{softmax}(\frac{\mathbf{Q}\mathbf{K}_o^T}{\sqrt{d}})\mathbf{V}_o + \text{softmax}(\frac{\mathbf{Q}\mathbf{K}_c^T}{\sqrt{d}})\mathbf{V}_c, \tag{4}$$

where $\mathbf{Q} = \mathbf{Z}\mathbf{W}_q, \mathbf{K}_o = f_o\mathbf{W}_k^o, \mathbf{V}_o = f_o\mathbf{W}_v^o, \mathbf{K}_c = f_c\mathbf{W}_k^c, \mathbf{V}_c = f_c\mathbf{W}_v^c$. $f_o$ represents the features of video frames extracted via the OpenCLIP image encoder, whereas $f_c$ denotes the video

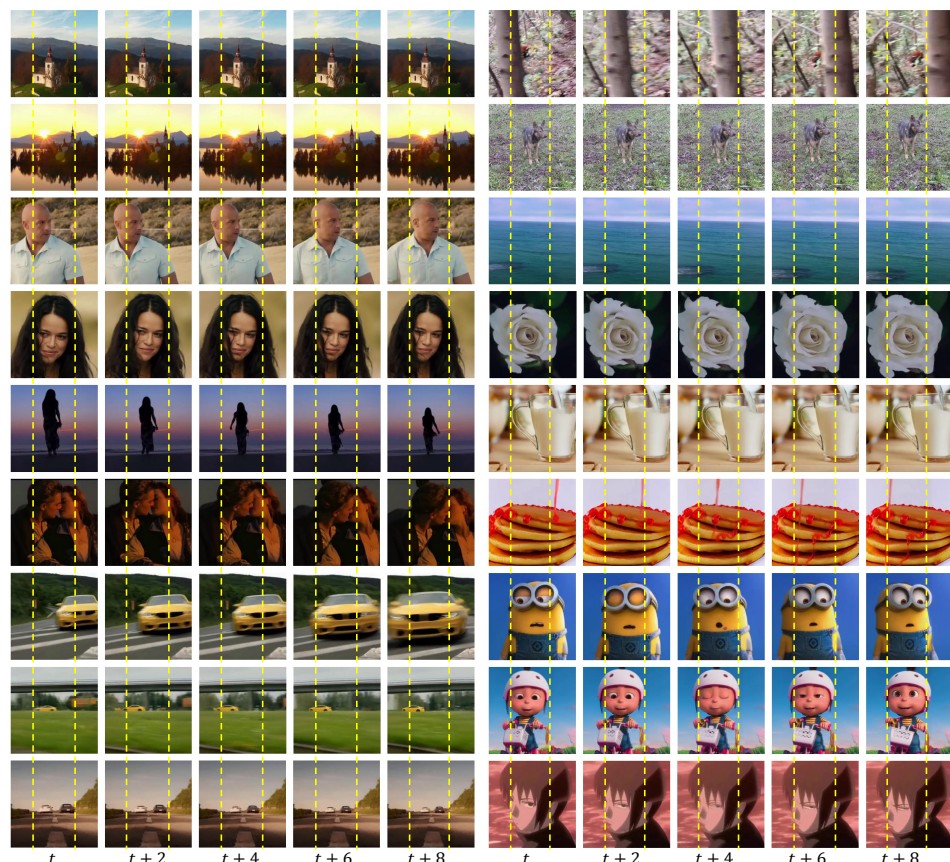

Figure 5: Qualitative results with mask ratio 50%. HERO demonstrates a robust ability to broaden diverse videos, encompassing landscapes, human figures (full-body, half-body, head-shot), swiftly moving vehicles, complex backgrounds, telefocus, nearfocus videos and even cartoon videos. Contents outside the yellow lines are outpainted. Central outpainting and horizontal Outpainting results can be seen in Fig. 9 and 10. Best viewed on screen with zoom.

features extracted from the OpenAI CLIP encoder. $\mathbf{Z}$ is the feature map from 3D-UNet and $\mathbf{W}_q$, $\mathbf{W}_k$, $\mathbf{W}_v$ are all learnable parameters.

**VAE features.** CLIP features are too coarse and do not contain texture information, which is not enough to describe the spatial information. The solution to remedy this problem is to utilize the features from the VAE model. The VAE model compresses images from pixel space into latent space and then restores them into pixel space with minimal loss. Thus, the VAE features contain rich texture information and can be used in outpainting tasks. Inspired by the image inpainting methods (Rombach et al., 2022; Razzhigaev et al., 2023), Features from the VAE are concatenated with the noise and a mask indicating the known and unknown areas, forming a new 9-channel input, which is then fed into the 3D-UNet.

## 3 EXPERIMENTS

### 3.1 DATASETS, BASELINES AND EVALUATION METRICS

**Datasets**. To validate the effectiveness of HERO, evaluations are conducted on SSV2 (Goyal et al., 2017), DAVIS (Caelles et al., 2019), YouTube-VOS (Xu et al., 2018). HERO is first trained and validated on the training and validation split of SSV2 respectively to strictly align with the MAGVIT (Yu et al., 2023). The training split of SSV2 contains 169K videos while the validation split contains about 24k. The SoTA results on DAVIS and YouTube-VOS are achieved by M3DDM (Fan et al., 2023) which is trained on an in-house 5M E-Commerce video data and evaluated on

Table 1: Video outpainting performance meaured by FVD on SSV2 dataset. A lower FVD score indicates better performance.

| Task | OPC↓ | OPV↓ | OPH↓ | AVG↓ |
|------|------|------|------|------|
| MAGVIT | 21.1 | 16.8 | 17.0 | 18.3 |
| M3DDM | 19.2 | 14.5 | 14.3 | 16.0 |
| HERO | **18.9** | **9.4** | **9.1** | **12.4** |

Table 2: Comparison with average adjacent frame similarity (AAFS) on DAVIS. The higher the value, the more stable the frames.

| Method | M3DDM | MOTIA | HERO |
|--------|-------|-------|------|
| AAFS↑ | 0.8650 | 0.8570 | **0.8768** |

DAVIS and YouTube-VOS. To align with M3DDM, we collect an equivalent magnitude of video data from the internet, train HERO on it and then evaluate HERO on the same data of DAVIS and YouTube-VOS with M3DDM (Fan et al., 2023).

**Baselines**. Our baselines include the following methods: the optical strategy based **Dehan** (Dehan et al., 2022), the masked-based **MAGVIT** (Yu et al., 2023), the diffusion-based **SDM** (Fan et al., 2023), **M3DDM** (Fan et al., 2023) and **MOTIA** (Wang et al., 2024a). Please refer to Sec. A for details of these methods.

**Evaluation Metrics**. For quantitative alignment with previous works, Mean Squared Error (**MSE**), Peak Signal To Noise Ratio (**PSNR**), structural similarity index measure (**SSIM**) (Wang et al., 2004), Learned Perceptual Image Patch Similarity (**LPIPS**) (Zhang et al., 2018) and Fréchet Video Distance (**FVD**) (Unterthiner et al., 2018) are adopted. The evaluation protocol is tightly aligned with M3DDM (Fan et al., 2023) and MAGVIT (Yu et al., 2023).

**Implementation details**. The weights of 3D-UNet and 3D-RefNet are initialized from Stable Diffusion 1.5. The weights of the vanilla motion modeling module is initialized from (Guo et al., 2024). The optical flow encoder is randomly initialized. The optimizer is AdamW (Loshchilov & Hutter, 2019), and the learning rate is constant at $1e-4$ and the weight decay at $1e-2$. The frame number is 16 across all experiments. The experiments are conducted on 16 NVIDIA A100 GPUs (80GB) with batch size 16 and gradient accumulation 16. HERO is trained for 9k steps on SSV2 and 13k steps for the self-collected video dataset (approximately 3 days). In the training phase, Central Outpainting (OPC), Vertical Outpainting (OPV), and Horizontal Outpainting (OPH) are trained using a multi-task approach. The video resolution is set to $256 \times 256$ on DAVIS and Youtube-VOS, $128 \times 128$ on SSV2 to maintain consistency with previous work. It takes 13 seconds to generate a 16-frame video with a resolution of $256 \times 256$ during the sampling stage.

### 3.2 QUALITATIVE RESULTS.

Fig. 5 demonstrates that HERO can expand various types of videos. In each video, only the middle 50% of the content is real content, while the content on the left and right sides is created by HERO.

### 3.3 COMPARISONS

**Quantitative comparison**. The quantitative comparison is first conducted on SSV2. As shown in Tab. 1, HERO achieves the best performance on all kinds of video outpainting tasks. When compared with MAGVIT, HERO demonstrates a decrease in average FVD by 5.9, utilizing an identical training set of SSV2. Moreover, in comparison with M3DDM, HERO also shows a drop in average FVD by 3.6, while training on merely 3.3% of M3DDM's training set amounting to 16.8K versus 500M.

Substantial research efforts such as M3DDM and MOTIA try to train video outpainting on large-scale datasets to boost performance. The HERO is also trained on 500M self-collected internet videos to deliver better performance and then is compared with them quantitatively on DAVIS and Youtube-VOS. As shown in Tab. 3, HERO achieves the best results on all five metrics for both datasets. These comparative analyses reveal that the proposed HERO architecture demonstrates clear superiority.

**Qualitative comparison**. As shown in Figure 6, the qualitative comparison primarily involves the latest algorithm M3DDM which is the most effective in current open source comparable models.

**Video stability comparison**. To better compare the stability of the video, the average adjacent frame similarity (AAFS) is used to describe temporal stability quantitatively. A higher AAFS value indicates greater video stability. The similarity is calculated using cosine similarity based on CLIP-V

Table 3: Video outpainting (OPV) performance on DAVIS and YouTube-VOS datasets. ↑ means "better when higher", and ↓ indicates "better when lower".

| Method | DAVIS dataset | | | | | YouTube-VOS dataset | | | | |
|---|---|---|---|---|---|---|---|---|---|---|
| Metric | PSNR↑ | SSIM↑ | MSE↓ | LPIPS↓ | FVD↓ | PSNR↑ | SSIM↑ | MSE↓ | LPIPS↓ | FVD↓ |
| Dehan | 17.96 | 0.6272 | 0.0260 | 0.2331 | 363.1 | 18.25 | 0.7195 | 0.02312 | 0.2278 | 149.7 |
| SDM | 20.02 | 0.7078 | 0.0153 | 0.2165 | 334.6 | 19.91 | 0.7277 | 0.01687 | 0.2001 | 94.81 |
| M3DDM | 20.26 | 0.7082 | 0.0149 | 0.2026 | 300.0 | 20.20 | 0.7312 | 0.01636 | 0.1854 | 66.62 |
| MOTIA | 20.36 | 0.7578 | — | 0.1595 | 286.3 | 20.25 | 0.7636 | — | 0.1727 | 58.99 |
| HERO | **20.82** | **0.7604** | **0.0143** | **0.1470** | **216.2** | **20.45** | **0.7699** | **0.01610** | **0.1608** | **56.87** |

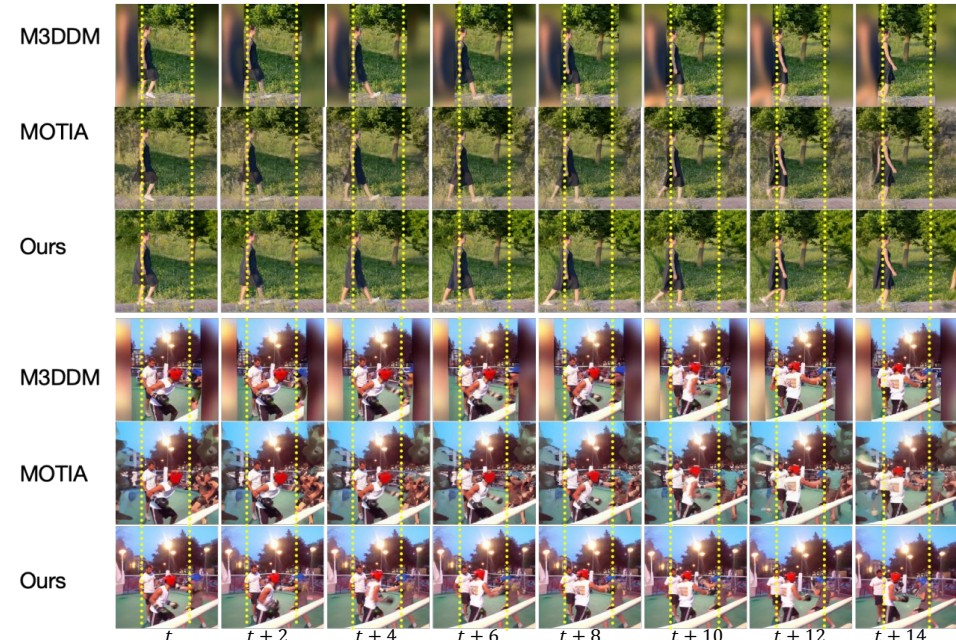

Figure 6: Qualitative comparisons results. Contents outside the yellow lines are outpainted. These **interesting video files** can be found in supplementary materials. Best viewed on screen with zoom.

features. These experiments are conducted on the DAVIS dataset, and the results are as Tab. 2. HERO has a relative improvement of 1.3% and 2.3% compared to M3DDM and MOTIA, respectively.

## 3.4 ABLATION STUDY

### 3.4.1 EFFECTIVE ON INTERPOLATED-BASED MOTION MODELING MODULE

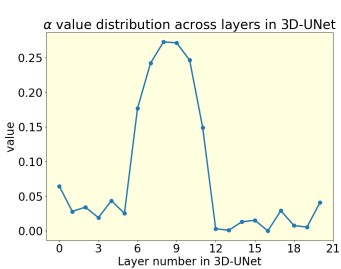

Figure 7: $\alpha$ across 3D-UNet layers.

Ablation studies are conducted on SSV2 test set with only 3k steps to reduce training time. As demonstrated in Tab. 4, the removal of the Interpolated-based Motion Modeling Module from 3D-RefNet results in a deterioration across all five metrics. Subsequently, its further removal from 3D-UNet aggravates this degradation, leading to further worsening of all five metrics. These findings underscore the indispensable role of the Interpolated-based Motion Modeling Module in HERO. In addition, the $\alpha$ values in each layer of 3D-UNet are illustrated. A clear trend can be seen from Fig. 7: $\alpha$ approaches to be close to 0 at shallow layers and takes on a value of about 0.3 at deeper layers. As discussed in Sec. 2.4, this phenomenon shows that features of adjacent frames tend to fuse at deep layers and remain independent at shallow layers.

Table 4: Ablation on the Interpolation-based Motion Modeling Module.

| 3D-UNet | 3D-RefNet | SSIM↑ | PSNR↑ | MSE↓ | LPIPS↓ | FVD↓ |
|---------|-----------|-------|-------|------|--------|------|
| ✓ | ✓ | **0.7908** | **19.83** | **0.0142** | **0.1709** | **10.03** |
| ✓ | ✗ | 0.7876 | 19.71 | 0.0150 | 0.1767 | 10.92 |
| ✗ | ✗ | 0.7853 | 19.59 | 0.0160 | 0.1831 | 11.01 |

Table 5: Ablation on Temporal Reference Module.

| Method | SSIM↑ | PSNR↑ | MSE↓ | LPIPS↓ | FVD↓ |
|--------|-------|-------|------|--------|------|
| Naïve Baseline | 0.7853 | 19.59 | 0.0160 | 0.1831 | 11.01 |
| + OFE w/ $t$ | 0.7935 | 20.08 | 0.0154 | 0.1792 | 10.64 |
| + OFE w/o $t$ | 0.7983 | 20.13 | 0.0141 | 0.1718 | 10.37 |
| w/o 3D-RefNet | 0.7715 | 18.43 | 0.0171 | 0.1976 | 14.52 |

### 3.4.2 EFFECTIVE ON TEMPORAL REFERENCE MODULE

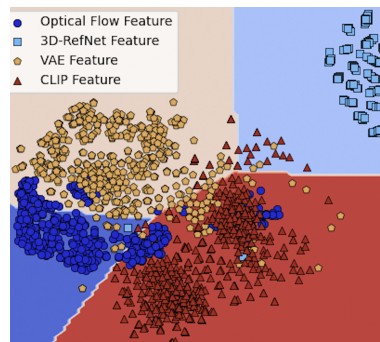

Figure 8: Distribution of complete spatial-temporal reference features.

The complete spatial-temporal reference features are illustrated in Fig. 8. It is clear that these reference features fully occupy the feature plane, as the cyan and blue regions are populated.

To ablate the Temporal Reference Module, a Naïve Baseline is first set up, and all ablation experiments are compared with it. The Interpolation-based Motion Modeling Module and the optical flow encoder are removed in the Naïve Baseline, and its results are shown in the first block in Tab. 5. It can be seen that when adding the optical flow encoder, the performance is improved, demonstrating the effectiveness of the optical flow encoder. "+ OFE encoder w/ $t$" synchronizes the OFE encoder's and 3D-UNet's timesteps during training, requiring iterative feature extraction for each timestep at inference. "+ OFE encoder w/o $t$" sets the OFE encoder's timestep to 0 during training for a single feature extraction at inference which is more effecient. Results in the second block shows that without timesteps, the optical flow encoder performs better and greatly reduces inference time with a single feature extraction. If the 3D-RefNet is removed, the FVD is increased from 11.01 to 14.52, which illustrates its indispensability.

## 4 RELATED WORK

### 4.1 DIFFUSION BASED VIDEO GENERATION

The structural principles of text-to-image models have had a significant influence on the development of text-to-video models following the successes of diffusion models in text-to-image tasks. Numerous studies (Esser et al., 2023; Ho et al., 2022; Hong et al., 2023; Khachatryan et al., 2023; Ma et al., 2024; Qi et al., 2023; Singer et al., 2023; Wu et al., 2023; Yang et al., 2023; Blattmann et al., 2023) have been conducted to augment text-to-image models with inter-frame attention mechanisms, aiming to facilitate the generation of videos. Some works achieve video generation by inserting temporal modules into text-to-image models. Video LDM (Blattmann et al., 2023) proposes multi-stage training to retain the prior knowledge of text-to-image models, training videos only on the temporal modules. AnimateDiff (Guo et al., 2024) uses a text-to-image model as the base generator, with an added motion modeling module to learn motion information. However, the motion modeling module is still a vanilla version and faces challenges in achieving stable video generation.

### 4.2 VIDEO OUTPAINTING

Currently, video outpainting technology is still not mature. Dehan (Dehan et al., 2022) proposed a background estimation technique that combines video object segmentation and video inpainting

methods, while temporal conherence is achieved through the integration of optical flow. However, in scenarios featuring complex camera movements and the exit of foreground objects from the frame, their performance frequently suffers. MAGVIT (Yu et al., 2023) introduced a versatile mask-based model designed for video generation that is also applicable to video outpainting tasks. It employs a 3D-VectorQuantized (3DVQ) tokenizer for video quantization and utilizes a transformer for conditional masked token modeling across multiple tasks. MAGVIT represents an inspirational effort, yet there is considerable room for improvement in its effectiveness. M3DDM (Fan et al., 2023) has designed an architecture based on the diffusion model, which is trained on massive datasets and achieves quite impressive results. However, there is a considerable proportion of bad cases with this approach, as shown in Figure 2b. The primary cause behind these issues is the insufficiency of reference information.

## 5 SOCIETAL IMPACTS, LIMITATIONS AND CONCLUSION

**Societal Impacts**. The proposed HERO is inherently harmless like many other AI technologies. Nevertheless, there exists the potential for its misuse, such as incorporation into applications with copyright issues, which could have negative effects on society. Hence, we advocate for the thoughtful and ethical application of HERO.

**Limitations**. (1) HERO has not specifically studied the long videos outpainting. When generating long videos, the coarse-to-fine generation strategy from M3DDM (Fan et al., 2023) or the recursive generation strategy from Hallo (Xu et al., 2024) can be employed. (2) The 3D-RefNet and the optical flow will take up more GPU memory. However, the two modules do not significantly increase the inference time as they do not require recurrent denoising and only need a single forward pass. (3) The method to integrate optical flow with OFE still has significant potential for enhancement.

**Conclusion**. This paper proposed a pioneering approach effectively harnessing the temporal modeling for diffusion-based outpainting (HERO) to handle the generated video quality problem. The Temporal Reference Module provides videl-level and motion features to assist the video generation, effectively addressing the limitations of VAE and CLIP. The Interpolation-based Motion Modeling Module utilizes adjacent frame relations to stabilize the frames with minimal modification to the network structure. Qualitative and quantitative experiments validate the superiority and robustness of HERO.

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

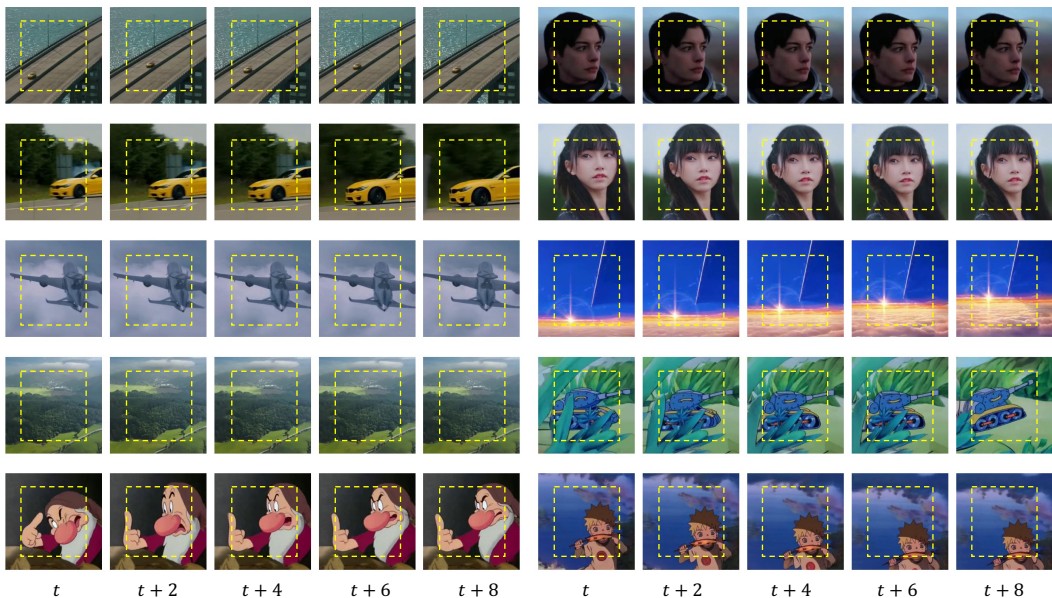

$$t \qquad t+2 \qquad t+4 \qquad t+6 \qquad t+8 \qquad t \qquad t+2 \qquad t+4 \qquad t+6 \qquad t+8$$

Figure 9: Qualitative results for OPC with a mask ratio of $50\%$. Contents outside the yellow lines are outpainted. Best viewed on screen with zoom.

## A    THE DETAILS OF BASELINES

The baseline includes: 1) **Dehan** (Dehan et al., 2022) develops a framework dedicated to the task of video outpainting. Their strategy involve differentiating between foreground and background elements, then estimating flow and background separately. These components are then integrated to produce a comprehensive output. 2) **MAGVIT** deployed mask modeling technology for the training of a transformer aimed at generating videos within the 3D Vector-Quantized (Esser et al., 2021) (van den Oord et al., 2017) space. 3) **SDM** model (Fan et al., 2023) utilizes the initial and terminal frames of a sequence as conditional inputs, which are integrated with contextual information at the inception layer of the network. This model has undergone training on video datasets, specifically WebVid (Bain et al., 2021) and an e-commerce dataset (Fan et al., 2023). 4) **M3DDM** (Fan et al., 2023) represents a pioneering approach to video outpainting, incorporating a masking strategy that enables the utilization of the original source video as masked conditions. Furthermore, it leverages global-frame features within cross-attention mechanisms to facilitate the accomplishment of comprehensive and extended information dissemination. The model underwent training utilizing two datasets containing a vast array of video data, specifically WebVid and e-commerce (Fan et al., 2023), and was fine-tuned on the corresponding datasets during evaluation. SDM can be considered a simplified version of M3DDM. 5) **MOTIA** (Wang et al., 2024a) employs spatial-aware insertion and noise travel to better harness the prior knowledge of the diffusion model as well as the video patterns in source videos.

## B    ADDITIONAL RESULTS

We show the results of Central Outpainting (OPC) and Horizontal Outpainting (OPH) with mask ratio $50\%$ in Figure 9 and  10. For OPC, the content on top,bottom,left and right sides of the video is created by HERO. For OPH, the top and bottom parts of the video are created by HERO.

## C    PROPORTION OF BAD CASES OF BASELINES

We manually viewed the output of M3DDM (Fan et al., 2023) on DAVIS and YouTube-VOS one by one, and count the ratio of generated video blur respectively and show them in Tab. 6

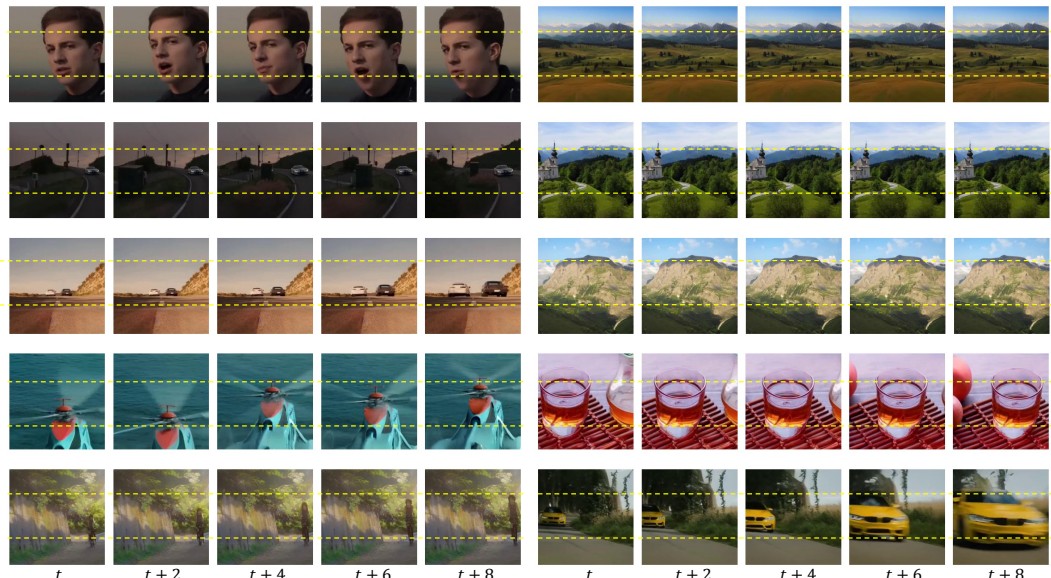

Figure 10: Qualitative results for OPH with a mask ratio of $50\%$. Contents outside the yellow lines are outpainted. Best viewed on screen with zoom.

Table 6: The proportion of bad cases with blurred details in M3DDM.

| Dataset | mask ratio = 0.666 | mask ratio = 0.25 |
|---|---|---|
| DAVIS dataset | 0.38 | 0.12 |
| YouTube-VOS dataset | 0.24 | 0.06 |

# D    ABLATION OF CLIP ENCODERS

The CLIP encoder is also ablated in Tab. 7. It shows that both CLIP encoders are useful to HERO.

Table 7: Ablation on CLIP encoders.

| Method | SSIM↑ | PSNR↑ | MSE↓ | LPIPS↓ | FVD↓ |
|---|---|---|---|---|---|
| only Open CLIP Encoder | 0.7840 | 19.52 | 0.0159 | 0.1853 | 11.99 |
| only OpenAI CLIP Encoder | 0.7836 | 19.47 | 0.0163 | 0.1849 | 12.01 |
| Both CLIPs | 0.7853 | 19.59 | 0.0160 | 0.1831 | 11.01 |

