# OpenReview forum: "HERO: Harnessing Temporal Modeling for Diffusion-Based Video Outpainting"
_ICLR.cc/2025/Conference — Submitted to ICLR 2025_

### Official Review · Reviewer_Mnkm · 2024-10-28

**Soundness:** 2
**Presentation:** 2
**Contribution:** 3
**Rating:** 5
**Confidence:** 3

**Summary:**

This paper proposes a novel approach for high-quality video outpainting. The temporal reference module provides a new perspective besides the spatial perspective, alleviating the inadequate temporal modeling problem. The proposed interpolation-based motion modeling module utilizes adjacent frame relations, enabling more stable results. Qualitative and quantitative experiments show that the proposed approach outperforms previous methods.

**Strengths:**

1. In addition to the spatial perspective, this paper gives insight into diffusion-based video outpainting from a temporal perspective, proposing corresponding solutions to remove temporal limitations from prior studies.

2. This paper innovatively introduces optical flow to enrich temporal information and strengthen the relationship between adjacent frames. Besides, the proposed $IM^3$ improves the stability of results in a low learning cost.

**Weaknesses:**

1. The parameters and computation for the network introduced in this paper are notable. Although its impacts on GPU memory and inference time are discussed in Lines 508-510, quantitative evaluation of parameters and computations should be demonstrated and compared with related methods.

2. The ablation results in Table 4 and Table 5 are insufficient to support that $IM^3$ is necessary for both 3D-RefNet and 3D-UNet (Line 471-472). Ablation experiments for 3D-UNet w/o $IM^3$ should be conducted.

3. Some minor errors: Check the spelling of “INTRODUTION” which leaves out the letter C (Line 047). The sentence on line 159 misses a period. There is an extra space on line 161. Check the citation format of the sentence on line 186. Missing a space in the sentence on line 782. The positions of Binary Mask and Optical Flow in Figure 3 are reversed.

**Questions:**

1. As stated in lines 155-156, the padded video is transformed into optical flow maps and then concatenated with binary masks, while on lines 209-211, it claimed that the dense optical flow of the input video is first estimated and the unknown regions will be filled with zero. It is confusing whether the input video is padded or not when calculating optical flow. Besides, if follow the statement on lines 155-156 that the input video is padded, the padding area will affect the generation of optical flow in known regions, making an influenced input.

2. Considering that the first and last frames only have one adjacent frame rather than two, does the lack of enhancement for these frames in the $IM^3$ module potentially result in poorer restoration quality for them?

---

> ### Author Response · Authors · 2024-11-24
> **Response to Reviewer Mnkm (Part1/2)**
>
> Thank you for carefully reading through HERO and providing thoughtful feedback.
> Your suggestions have significantly contributed to enhancing the quality of the paper.
> We trust that the clarifications below will resolve your concerns.
>
>
> **\[Q1. The model parameters and computation of HERO.\]**
>
>
> Thank you for raising the insightful point regarding model parameters and computational cost. We deeply appreciate it and have taken efforts to conduct thorough measurements.
>
> - In addition to the diffusion-based video outpainting baselines M3DDM[1] and MOTIA[2], we have also included two notable video generation methods, Animatediff[3] and AnimateAnyone[5], in our evaluation. These methods, whose network designs are widely referenced in the video generation field, share many similarities with M3DDM, MOTIA, and HERO.
> - We observed that many prior works[1,3,5] based on diffusion models **lack comprehensive statistics on parameters and computational requirements**. To address this gap, **we set up an environment to align benchmarks and fill this shortcoming in the community**. Furthermore, We have published our evaluation protocols, including the evaluation tools, video resolution details, hardware specifications, and metrics, to establish a solid benchmark. **This paves a way for future research in the community to build upon**. For our evaluations,
> 	- **Evaluation Tool.** We used the ``calflops`` tool from GitHub to ensure standardized assessments.
> 	- **Video Resolution.** Video outpainting was conducted with a resolution of 256x256 to maintain fairness and consistency, and all other parameters adhered to the default settings of the open-source methods.
> 	- **Device.** Model speed tests were performed on an NVIDIA A100 GPU.
> 	- **Metrics.** We measured dimensions such as computational cost, parameter count, inference time, and peak memory usage.
> 	- **Formats.** We report the mean and variance of the inference times based on 10 runs following 3 warmup iterations.
> - **This table below will be included in the main text as a foundational resource, serving as a solid reference for future researchers.  This will encourage future papers on to honestly report their parameter counts and computational costs, fostering healthy development within both the research community and the industry.**
>
> The statistical results are shown in the table below.
>
> |               | TFLOPS↓ | Parameter Size(M)↓ | Inference Time↓ | Peak Memory↓ |
> | ------------- | ------- | ------------------ | --------------- | ------------ |
> | AnimateDiff   | 250.36  | 1687.6             | 9.90±0.02s      | 15.07G       |
> | AnimateAnyone | 255.06  | 1906.4             | 12.57±0.04s     | 12.9G        |
> | M3DDM         | 245.44  | 1109.57            | 21.27±0.09 s    | 13.96G       |
> | MOTIA         | 259.44  | 1692.52            | 10.53±0.01s     | 14.28G       |
> | HERO          | 259.99  | 2399.09            | 13.00±0.23 s    | 31.05G       |
>
> From the table above, it can be observed that:
> - **HERO significantly increases parameter counts.** As noted in the **Limitations section**  of this paper, HERO does have shortcomings in terms of parameter count and memory usage. We have **truthfully**  reported these limitations to ensure users are aware of these factors. As shown in the table, HERO requires more resources, with a 41.74% increase in parameters and approximately a two-fold increase in memory.
> - **The increase in computational cost is trivial.** However, despite the significant increase in parameters, these additional parameters are involved in the computation only once during video outpainting and **do not require iterative processing** during the denoising stage, thus having a **negligible impact** on computational cost. In terms of computational cost (TFLOPS), HERO only increases by **0.2%** compared to the previous state-of-the-art (SoTA), MOTIA.
> - **The inference time.** Consequently, the inference time of our method remains comparable to MOTIA, with only a modest 23% increase. For video generation tasks, 13s is still within a reasonable range.
> - **Performance trade-off**. Nevertheless, HERO demonstrates significant improvements in both quantitative and qualitative results, as well as in newly added user studies. The trade-off in performance for enhanced outcomes is justified.
> - **Mitigation measures.** In situations with limited memory resources, if you want to use HERO, you can extract the features of modules like OFE and 3D-RefNet offline and use them during the denoising process, which can save up to half of the memory.
> - It is important to specifically note that M3DDM employs a coarse-to-fine strategy, which improves frame quality but requires generating multiple video frames, resulting in higher inference time.
>
> We hope the responses can address your concerns.

---

> ### Author Response · Authors · 2024-11-24
> **Response to Reviewer Mnkm (Part2/2)**
>
> **\[Q2. The necessity of $\text{IM}^3$.\]**
>
> The necessity of  $\text{IM}^3$ can be thoroughly validated in Table 4.
> The checkmark symbol (√) indicates that the corresponding module includes  $\text{IM}^3$, while the cross (×) represents that it does not include  $\text{IM}^3$.
>
> | 3D-UNet | 3D-RefNet | SSIM↑  | PSNR↑ | MSE↓   | LPIPS↓ | FVD↓  |
> | ------- | --------- | ------ | ----- | ------ | ------ | ----- |
> | √ | √   | 0.7908 | 19.83 | 0.0142 | 0.1709 | 10.03 |
> | √ | ×  | 0.7876 | 19.71 | 0.0150 | 0.1767 | 10.92 |
> | ×| ×  | 0.7853 | 19.59 | 0.0160 | 0.1831 | 11.01 |
> As can be seen from the table above,
> - The comparison between Row 1 and Row 2 shows that $\text{IM}^3$ is essential for 3D-RefNet.
> - Similarly, the comparison between Row 2 and Row 3 demonstrates the necessity of $\text{IM}^3$ for 3D-UNet.
>
> Regarding Table 5, the last row was mistakenly included due to an oversight during remote collaboration, incorporating an outdated result. This row has been removed in the revised manuscript.
>
>
> **\[Q3. The literal details of HERO.\]**
>
> We appreciate your attention to detail and thank you for carefully reviewing our manuscript.
> All typos have been corrected, and the figure labeling error in Figure 3 has been addressed as you suggested in the newly submitted manuscript.
>
>
> **\[Q4. The order of optical flow estimation and video padding.\]**
>
> Thank you again for your insightful review to make HERO more solid.
> We have restructured the description of this process.
> We will updated the manuscript to make this distinction clearer.
>
> - The input video is first estimated for the optical flow without any padding.
> - After this step, the video is padded to the target size to ensure it aligns with the network’s input requirements.
> - The padding itself does not affect the optical flow estimation, as the optical flow is calculated based on the original, unpadded video.
>
>
>
> **\[Q5. The limitations on boundary frames.\]**
>
> Thank you for your thoughtful question regarding the first and last frames in the $\text{IM}^3$ module.
> You are correct that boundary frames have fewer adjacent frames than middle frames, which could slightly affect their restoration quality.
>
> - We plan to explore targeted improvements for boundary frames in future work.
> - However, we believe the $\text{IM}^3$ module effectively enhances most frames by leveraging available adjacent frames for fusion.
> - What is more, in autoregressive long-video generation, this limitation is naturally resolved, as boundary frames eventually become middle frames in subsequent steps and receive enhancement apart from the global first and last frames..
>
>
>
> [1] Fan, F., Guo, C., Gong, L., Wang, B., Ge, T., Jiang, Y., ... & Zhan, J. (2023, October). Hierarchical masked 3d diffusion model for video outpainting. In _Proceedings of the 31st ACM International Conference on Multimedia_ (pp. 7890-7900).
>
> [2] Wang, F. Y., Wu, X., Huang, Z., Shi, X., Shen, D., Song, G., ... & Li, H. (2025). Be-your-outpainter: Mastering video outpainting through input-specific adaptation. In _European Conference on Computer Vision_ (pp. 153-168). Springer, Cham.
>
> [3] Guo, Y., Yang, C., Rao, A., Liang, Z., Wang, Y., Qiao, Y., ... & Dai, B. (2023). Animatediff: Animate your personalized text-to-image diffusion models without specific tuning. International Conference on Learning Representations (ICLR) 2024.
>
> [4] Zhang, L., Rao, A., & Agrawala, M. (2023). Adding conditional control to text-to-image diffusion models. In _Proceedings of the IEEE/CVF International Conference on Computer Vision_ (pp. 3836-3847).
>
> [5] Hu, L. (2024). Animate anyone: Consistent and controllable image-to-video synthesis for character animation. In _Proceedings of the IEEE/CVF Conference on Computer Vision and Pattern Recognition_ (pp. 8153-8163).

---

### Official Review · Reviewer_kZW5 · 2024-10-29

**Soundness:** 3
**Presentation:** 2
**Contribution:** 2
**Rating:** 5
**Confidence:** 4

**Summary:**

This paper describes a method called HERO, which solves the problems of blurred details, local distortion, and temporal instability common in existing methods by introducing two key components: a temporal reference module and an interpolation-based motion modeling module. The former provides reference features beyond the spatial dimension, stabilising the generated frames through the relationship between neighboring frames.

**Strengths:**

HERO has shown qualitative and quantitative results in several benchmark tests that outperform existing methods.
The HERO design consists of two independent but complementary modules that can be flexibly applied to different video generation tasks.

**Weaknesses:**

The paper points out issues with existing diffusion model-based video episodic processing, like detail blurring, local distortions, and temporal instability. However, it doesn't dive deep into what’s causing these problems. It claims inadequate temporal modeling is the main issue but doesn’t explain why current methods can’t tackle these challenges effectively.
There’s a noticeable lack of innovation in the paper. The time-referenced module and the interpolation-based motion modeling module in HERO mainly combine existing models like VAE, 3D-RefNet, 3D-UNet, and other diffusion models that are already well-known in this field.
While HERO shows promising results in various benchmarks, the experimental section doesn’t properly validate important aspects like model parameters and speed. These factors are crucial for a full evaluation of how well the model performs.

**Questions:**

The author’s description of the model structure in section 2.2 doesn’t align with what’s shown in Figure 3. This is particularly evident in the Temporal Reference Perspective section, where it’s unclear why the results from Optical Flow Enc are handed off to the second half of 3D-UNet.
In Figure 3, it’s unclear why the Spatial Reference Perspective shows CLIP pointing to both the front and back parts of 3D-UNet. This could use some clarification.

---

> ### Author Response · Authors · 2024-11-24
> **Response to Reviewer kZW5 (Part1/3)**
>
> We sincerely thank you for taking the time to carefully read HERO and share your thoughtful insights.
> Your feedback have contributed to strengthening the quality of HERO.
> We hope the following clarifications address your concerns.
>
>
> **\[Q1. The innovation of HERO. \]**
> - **The contributions of HERO.** As far as we know, HERO is the first work to hit the issue of insufficient temporal modeling in existing methods and explicitly harness temporal modeling to improve video outpainting performance.
> 	- **Video editing differs from image editing.** The temporal modeling is crucial for the success of diffusion models in this domain. However, most existing methods lack dedicated temporal design. Any local distortion in a single frame will propagate throughout the video during the UNet denoising process, making the video difficult to view. This also limits UNet’s spatial awareness due to the lack of temporal awareness.
> 	- HERO systematically propose a temporal reference module, including 3D-RefNet, Optical Flow Encoder, and the $\text{IM}^3$ module (Section 2.3). The effectiveness of our method is validated both qualitatively and quantitatively on SSV2, DAVIS, and YouTube-OS datasets.
> 	- The results, as shown in Figure 2(c) and Figure 8, further demonstrate the superiority of our approach. Specifically, Figure 2(c) illustrates that methods like M3DDM and MOTIA, which rely on CLIP and VAE, fail to fully utilize the feature plane. In contrast, Figure 8 shows that with our proposed reference modules, the reference features fully occupy the feature plane, clearly demonstrating the effectiveness of our method.
> 	- The ablation studies in Section 3.4 strongly support our claims regarding the advantages of the proposed modules.
>
> - **The baseline method M3DDM suffer from inadequate temporal modeling**.
> 	- M3DDM consists of two modules: the 3D-UNet and CLIP (as shown in Figure 3 in [1]). The UNet, originally designed for static images, inherently lacks any temporal understanding.
> 	- To address the limitations of UNet, M3DDM integrates a basic motion modeling module to form the 3D-UNet, leveraging a vanilla temporal transformer initially introduced in Animatediff [3]. However, M3DDM’s temporal modeling relies solely on pixel-wise attention across frames, which is simplistic and offers limited effectiveness. This is precisely the area that HERO aims to improve upon by proposing the $\text{IM}^3$ module, which is validated in Section 3.4.1.  **As a result, M3DDM's temporal modeling is inadequate**.
> 	- As for the CLIP module, features extracted through CLIP are independent of each other and there is no interaction between frames.  This is why HERO proposed the temporal modules in Section 2.3.1 and 2.3.2.  What is more, Figure 2(c) illustrates that features from CLIP and VAE, fail to fully utilize the feature plane. In contrast, Figure 8 shows that with our proposed reference modules, the reference features fully occupy the feature plane, clearly demonstrating the effectiveness of our method. **Therefore, M3DDM demonstrates weaknesses in temporal modeling**.
> 	- M3DDM adopts a coarse-to-fine strategy to ensure temporal consistency across generated frames. However, it is only effective in generating long videos and has no effect on the video snippet. In contrast, HERO incorporates sufficient temporal modeling for the generation of each video snippet **Accordingly, M3DDM suffer from  inadequate temporal modeling**.
> 	- As a result, M3DDM performs worse than HERO in both qualitative and quantitative evaluations.
>
> - **The baseline method MOTIA also struggles with a lack of proper temporal modeling**.
> 	- The network structure of MOTIA (as shown in Figure 3 of [2]) is largely similar to that of M3DDM, and therefore, **it also inherits all the aforementioned shortcomings related to inadequate temporal modeling**.
> 	- MOTIA uses the LoRAs during training and introduces Pattern-Aware Outpainting in Section 4.4[2]. It increases insertion weights near known areas to leverage the learned patterns while decreasing it in distant regions to rely more on the original generative capacity of the diffusion model. This is a good idea, **but this is only spatial modeling, not temporal modeling**.
> 	- MOTIA also introduce the noise regret mechanism in denoising process, which is a technique to merge noisy states from  from known and unknown regions. Although this strategy may improve frame quality, **it still does not model the temporal issues**.
> 	- Therefore, MOTIA performs better than M3DDM but still **overlooks the significant improvement that temporal modeling could bring**. This is exactly what HERO contributes.
>
> We hope the clarifications can resolve your concerns.

---

> > ### Comment · Reviewer_kZW5 · 2024-11-26
> > **Similarity with existing works.**
> >
> > The motion modelling module of the 3D-UNet described is just an optical flow module, which has already appeared many times in relation to other tasks, as follows.
> > [1] Ni H, Shi C, Li K, et al. Conditional image-to-video generation with latent flow diffusion models[C]//Proceedings of the IEEE/CVF conference on computer vision and pattern recognition. 2023: 18444-18455.
> > [2] Liang J, Fan Y, Zhang K, et al. MoVideo: Motion-Aware Video Generation with Diffusion Model[C]//European Conference on Computer Vision. Springer, Cham, 2025: 56-74.
> > [3] Lee M, Cho S, Shin C, et al. Video Diffusion Models are Strong Video Inpainter[J]. arXiv preprint arXiv:2408.11402, 2024.

---

> > > ### Author Response · Authors · 2024-11-27
> > > **HERO’s Contributions Are Overlooked: Optical Flow Doesn't Mean Similar Work.**
> > >
> > > Thank you for taking the time to review and provide feedback on our work.
> > >
> > > 1. The temporal modeling of HERO is comprehensively detailed in Sections 2.3-2.4, highlighting its **three core components**: 3D-RefNet, Optical Flow Encoder (OFE), and $\text{IM}^3$. Therefore, the assertion that “the motion modeling module of the 3D-UNet is just an optical flow module” is overly simplistic.
> > >
> > > 2. HERO firstly introduces 3D-RefNet for video-level reference features and **upgrades the naive motion modeling module to the Interpolation-based Motion Modeling module ($\text{IM}^3$)** for enhanced stability in video generation. The effectiveness of these modules has been validated in Tables 4-5.  Additionally, OFE was firstly introduced to the video outpainting because we found that optical flow information is crucial for this task.
> > > HERO’s contributions **extend far beyond the inclusion of OFE**; please refer to our contribution summary in Lines 117-124 for a detailed overview.
> > >
> > > 3. HERO is **the first work to address the limitations of temporal modeling** in existing diffusion methods and explicitly integrates temporal modeling to improve video outpainting performance.
> > > HERO reflects our innovation in advancing diffusion models from image generation to the challenging domain of video editing.
> > >
> > > 4. The use of optical flow in our work should not be dismissed as lacking innovation simply just because it has been applied by others in different tasks.
> > >
> > > We kindly ask the reviewer to consider HERO’s broader contributions carefully.

---

> ### Author Response · Authors · 2024-11-24
> **Response to Reviewer kZW5 (Part2/3)**
>
> **\[Q2. The model parameters and speed of HERO.\]**
>
> Thank you for raising the insightful point regarding model parameters and computational cost. We deeply appreciate it and have taken efforts to conduct thorough measurements.
>
> - In addition to the diffusion-based video outpainting baselines M3DDM[1] and MOTIA[2], we have also included two notable video generation methods, Animatediff[3] and AnimateAnyone[5], in our evaluation. These methods, whose network designs are widely referenced in the video generation field, share many similarities with M3DDM, MOTIA, and HERO.
> - We observed that many prior works[1,3,5] based on diffusion models **lack comprehensive statistics on parameters and computational requirements**. To address this gap, **we set up an environment to align benchmarks and fill this shortcoming in the community**. Furthermore, We have published our evaluation protocols, including the evaluation tools, video resolution details, hardware specifications, and metrics, to establish a solid benchmark. **This paves a way for future research in the community to build upon**. For our evaluations,
> 	- **Evaluation Tool.** We used the ``calflops`` tool from GitHub to ensure standardized assessments.
> 	- **Video Resolution.** Video outpainting was conducted with a resolution of 256x256 to maintain fairness and consistency, and all other parameters adhered to the default settings of the open-source methods.
> 	- **Device.** Model speed tests were performed on an NVIDIA A100 GPU.
> 	- **Metrics.** We measured dimensions such as computational cost, parameter count, inference time, and peak memory usage.
> 	- **Formats.** We report the mean and variance of the inference times based on 10 runs following 3 warmup iterations.
> - **This table below will be included in the main text as a foundational resource, serving as a solid reference for future researchers.  This will encourage future papers on to honestly report their parameter counts and computational costs, fostering healthy development within both the research community and the industry.**
>
> The statistical results are shown in the table below.
>
> |               | TFLOPS↓ | Parameter Size(M)↓ | Inference Time↓ | Peak Memory↓ |
> | ------------- | ------- | ------------------ | --------------- | ------------ |
> | AnimateDiff   | 250.36  | 1687.6             | 9.90±0.02s      | 15.07G       |
> | AnimateAnyone | 255.06  | 1906.4             | 12.57±0.04s     | 12.9G        |
> | M3DDM         | 245.44  | 1109.57            | 21.27±0.09 s    | 13.96G       |
> | MOTIA         | 259.44  | 1692.52            | 10.53±0.01s     | 14.28G       |
> | HERO          | 259.99  | 2399.09            | 13.00±0.23 s    | 31.05G       |
>
> From the table above, it can be observed that:
> - **HERO significantly increases parameter counts.** As noted in the **Limitations section**  of this paper, HERO does have shortcomings in terms of parameter count and memory usage. We have **truthfully**  reported these limitations to ensure users are aware of these factors. As shown in the table, HERO requires more resources, with a 41.74% increase in parameters and approximately a two-fold increase in memory.
> - **The increase in computational cost is trivial.** However, despite the significant increase in parameters, these additional parameters are involved in the computation only once during video outpainting and **do not require iterative processing** during the denoising stage, thus having a **negligible impact** on computational cost. In terms of computational cost (TFLOPS), HERO only increases by **0.2%** compared to the previous state-of-the-art (SoTA), MOTIA.
> - **The inference time.** Consequently, the inference time of our method remains comparable to MOTIA, with only a modest 23% increase. For video generation tasks, 13s is still within a reasonable range.
> - **Performance trade-off**. Nevertheless, HERO demonstrates significant improvements in both quantitative and qualitative results, as well as in newly added user studies. The trade-off in performance for enhanced outcomes is justified.
> - **Mitigation measures.** In situations with limited memory resources, if you want to use HERO, you can extract the features of modules like OFE and 3D-RefNet offline and use them during the denoising process, which can save up to half of the memory.
> - It is important to specifically note that M3DDM employs a coarse-to-fine strategy, which improves frame quality but requires generating multiple video frames, resulting in higher inference time.
>
> We hope the responses can address your concerns.

---

> > ### Comment · Reviewer_kZW5 · 2024-11-26
> >
> > Thank you for your reply, I have no problem with this.

---

> ### Author Response · Authors · 2024-11-24
> **Response to Reviewer kZW5 (Part3/3)**
>
> **\[Q3. The methods for connecting OFE and CLIP to UNet.\]**
>
> The 3D-UNet consists of two parts: the front part is the encoder, and the back part is the decoder. The encoder is responsible for extracting abstract features, while the decoder is used for generation.
> - **The Optical Flow Encoder only connects to the decoder**.
> 	- Optical Flow Encoder serves as an alternative encoder specifically designed to extract motion-specific information, running parallel to the 3D-UNet encoder. Therefore, its output is directed solely to the decoder, creating a shared-decoder paradigm with multiple encoders.
> 	- This design also follows practices similar to ControlNet[4], where conditions are injected into later stages to preserve the robustness of the pretrained encoder and avoid disrupting global feature extraction.
> - **CLIP guidance is applied at both the encoder and decoder**.
> 	- As described in Section 2.5, CLIP extracts global semantic features, providing consistent guidance throughout the whole process.
> 	- CLIP embeddings are injected into both the encoder and decoder of the 3D-UNet, ensuring semantic alignment during feature extraction and refinement during reconstruction.
> 	- This design is consistent with most diffusion-based controllable generation models [1, 2, 3, 5].
>
> We hope the explanation can answer your questions.
>
>
> [1] Fan, F., Guo, C., Gong, L., Wang, B., Ge, T., Jiang, Y., ... & Zhan, J. (2023, October). Hierarchical masked 3d diffusion model for video outpainting. In _Proceedings of the 31st ACM International Conference on Multimedia_ (pp. 7890-7900).
>
> [2] Wang, F. Y., Wu, X., Huang, Z., Shi, X., Shen, D., Song, G., ... & Li, H. (2025). Be-your-outpainter: Mastering video outpainting through input-specific adaptation. In _European Conference on Computer Vision_ (pp. 153-168). Springer, Cham.
>
> [3] Guo, Y., Yang, C., Rao, A., Liang, Z., Wang, Y., Qiao, Y., ... & Dai, B. (2023). Animatediff: Animate your personalized text-to-image diffusion models without specific tuning. International Conference on Learning Representations (ICLR) 2024.
>
> [4] Zhang, L., Rao, A., & Agrawala, M. (2023). Adding conditional control to text-to-image diffusion models. In _Proceedings of the IEEE/CVF International Conference on Computer Vision_ (pp. 3836-3847).
>
> [5] Hu, L. (2024). Animate anyone: Consistent and controllable image-to-video synthesis for character animation. In _Proceedings of the IEEE/CVF Conference on Computer Vision and Pattern Recognition_ (pp. 8153-8163).

---

> > ### Comment · Reviewer_kZW5 · 2024-11-26
> >
> > Thank you for your response. As you pointed out, the novelty of this architecture could be further refined. Such a strong model will be more robust for most tasks, rather than proposing new features for the task.

---

> ### Author Response · Authors · 2024-11-27
> **Optical Flow Encoder is Specifically Designed to be Lightweight.**
>
> Thank you for your reply.
>
> In HERO, the Optical Flow Encoder (OFE) is **specifically designed to be lightweight**. Specifically,
> - Its structure is inspired by ControlNet, with only half the parameter count of 3D-UNet.
> - During inference, it requires only a single feature extraction and does not participate in the iterative denoising process, making its computational cost negligible (~1%).
>
> We hope the responses can address your concerns.

---

### Official Review · Reviewer_ddaE · 2024-11-04

**Soundness:** 3
**Presentation:** 2
**Contribution:** 2
**Rating:** 6
**Confidence:** 5

**Summary:**

# Summary

The paper proposes a diffusion-based video outpainting framework. The proposed framework designs a temporal reference module to better model the spatiotemporal representations and designs a interpolation-based motion modelling module for better temporal consistency. The proposed framework outperforms the selected video outpainting baselines

**Strengths:**

# Strengths

- The proposed modules both seems reasonable and technically sound
- Extensive experiments demonstrate the effectiveness of the proposed framework

**Weaknesses:**

# Weaknesses

- In Sec 2.1, the statements about denoising UNet seem incorrect; also DDPM is not a deterministic sampling process
- Considering there are no ground truth results for video outpainting, the usage of pixel-based metrics like MSE/PSNR/SSIM does not necessarily reflect the actual performance
- Diffusion models can generate diverse results with the same input conditions and different random seeds. The paper does not mention how results are selected are comparison. It might be better to repeat the experiments across different seeds to report mean+std and also report the success rate for each generation
- The paper does not mention if prompts were used in the evaluations
- The selected evaluation metrics do not necessarily reflect the actual visual quality and temporal consistency of generated videos and human evaluations remain the most reliable measure. User study should be used for comparisons
- We can still observe local jitting artifacts/object distortions in the outpainted regions (in supplementary videos)
- The paper does not evaluate the video outpainting on complicated scenes like occlusions, object re-appearance, fast motions.

# Other comments (not weaknesses)

- In Fig 3, the captions for "Binary Mask" and "Optical Flow" are incorrectly positioned
- The overview (Fig 3) lacks clarity, which might need a better visualization of the shared UNet

**Questions:**

Please refer to the weaknesses section

---

> ### Author Response · Authors · 2024-11-24
> **Response to Reviewer ddaE (Part1/3)**
>
> Thank you for dedicating time to review HERO and providing such thoughtful feedback.
> Your comments have contributed significantly to improving the quality of the paper.
> We hope the clarifications below will address your concerns.
>
>
> **\[Q1. The statements about denoising UNet.\]**
>
> We have carefully revised the relevant sections to ensure factual accuracy and clarity, addressing all concerns you raised. We greatly appreciate your feedback, which has helped us improve the rigor of our work.
> - **Diffusion Process Description:** We clarified that the perturbation is based on a predefined noise schedule rather than stepwise diffusion. The revised version now states: *During training, the image latent $\mathbf{z}_0$ is perturbed into a noisy latent $\mathbf{z}_t$ at a specified time step $t$ using a predefined noise schedule. The denoising UNet is trained to predict the added noise, conditioned on $t$ and other inputs.*
> - **DDPM and DDIM Sampling Process:** We corrected the description of DDPM and DDIM. DDPM is a probabilistic sampling process while DDIM is a deterministic sampling processes. The updated version reads: *During inference, $\mathbf{z}^\prime_T$ is sampled from a random Gaussian distribution at timestep $T$ and progressively denoised to $\mathbf{z}^\prime_0$ using a guided sampling process (*e.g.*, DDPM, DDIM).*
> - **Noise Prediction:** The explanation of noise prediction by the denoising UNet has been refined for clarity. The revised text states: *In each iteration, the denoising UNet predicts the noise component in the latent variable for a given timestep $t$.*
> - **Role of VAE:** We revised the description of the VAE’s role to better reflect its functionality. The updated version now states: *The encoder $\mathcal{E}$ of VAE transforms an image from pixel space into a low-dimensional latent space, reducing computational complexity while also providing a probabilistic representation essential for the diffusion process.*
>
> We hope these modifications address your concerns and improve the clarity and accuracy of our manuscript.
>
>
> **\[Q2. Limitations of pixel-based metrics.\]**
>
> Thank you for pointing out the limitations of pixel-based metrics.
> - While we acknowledge their limitations, MSE, PSNR, and SSIM are widely used in the video generation and editing community[1,2,5,6], serving as standardized benchmarks for quantitative evaluation.
> - In addition to pixel-based metrics, we recognize the importance of subjective evaluations. we conducted qualitative comparisons to assess visual quality and temporal consistency. Specifically, Figure 9 presents our qualitative results, Figures 1, 5, 6, and 10 demonstrate the performance of HERO under different outpainting settings, and comparative mp4 videos are provided in the supplementary materials.
> - To further demonstrate the superiority of HERO, we have supplemented our work with user studies in **\[Q5. User studies\]**, and the results are included in the responses below.
> - In future work, we plan to explore perceptual metrics and more sophisticated evaluation protocols tailored to video outpainting.
>
> **\[Q3. The mean±std evaluation.\]**
>
> Thank you for your thoughtful feedback and we greatly value and appreciate your rigorous approach.
> - Video generation is highly resource-intensive, particularly when evaluating a large dataset like SSV2 with 27k videos. Given the computational demands, it was challenging to test multiple random seeds within a short timeframe.
> - To address this, we conducted experiments on a smaller dataset, DAVIS, to evaluate the impact of different random seeds. Specifically, we tested three different random seeds and calculated the metrics’ mean and variance.
> - In the table below, _HERO (original)_ refers to the results from the original paper, while _HERO (new)_ represents the averaged metrics from three independent evaluations with different random seeds. The results show that HERO consistently maintains its state-of-the-art performance.
>
>
> |                | PSNR↑          | SSIM↑             | MSE↓              | LPIPS↓            | FVD↓           |
> | -------------- | -------------- | ----------------- | ----------------- | ----------------- | -------------- |
> | Dehan          | 17.96          | 0.6272            | 0.0260            | 0.2331            | 363.1          |
> | SDM            | 20.02          | 0.7078            | 0.0153            | 0.2165            | 334.6          |
> | M3DDM          | 20.26          | 0.7082            | 0.0149            | 0.2026            | 300.0          |
> | MOTIA          | 20.36          | 0.7578            | —                 | 0.1595            | 286.3          |
> | HERO(original) | **20.82**      | **0.7604**        | **0.0143**        | **0.1470**        | **216.2**      |
> | HERO(new)      | **20.83±0.10** | **0.7606±0.0007** | **0.0142±0.0001** | **0.1507±0.0036** | **226.0±10.4** |

---

> ### Author Response · Authors · 2024-11-24
> **Response to Reviewer ddaE (Part2/3)**
>
> - We have also included the individual results from each of the three evaluation rounds for your reference.
>
> |        | PSNR↑ | SSIM↑  | MSE↓   | LPIPS↓ | FVD↓   |
> | ------ | ----- | ------ | ------ | ------ | ------ |
> | round1 | 20.82 | 0.7604 | 0.0143 | 0.147  | 216.2  |
> | round2 | 20.93 | 0.7614 | 0.0141 | 0.1541 | 236.85 |
> | round3 | 20.74 | 0.76   | 0.0143 | 0.151  | 224.85 |
>
> - HERO rarely fails to generate videos, though some bad cases exist. These are discussed in detail in **Q5: Bad Case Study**.
>
>
>
> **\[Q4. The use of text prompts.\]**
>
> As illustrated in Figure 3, HERO does not utilize text prompts.
> HERO relies on the provided video references, which serve as sufficient visual prompts to generate plausible outpainting content.
>
>
> **\[Q5. User studies.\]**
>
> Thank you for highlighting the value of human evaluations, which we agree are essential for assessing the perceived quality of video outpainting.
> The human evaluations are conducted as follows.
> - **Participant Demographics:** The user study involved 12 participants, consisting of 1 females and 11 males. The group had diverse educational backgrounds, including 2 Ph.D. holders, 8 master’s degree holders, and 2 bachelor’s degree holders, with participants from at least two different cities, reflecting varied growth environments.
> - **Experimental Setup:** A total of 40 videos were randomly selected from the DAVIS dataset for evaluation. Each video was masked by removing 25% of the content from both the left and right sides, resulting in a total of 50% masked video content.
> - **Comparison Methods:** The performance of three methods was compared: M3DDM, MOTIA, and HERO.
> - **Blind Testing:** We ensured that the participants were unaware of which method each video came from, and the arrangement of the videos from the three methods was shuffled for each round of scoring.
> - **Evaluation Criteria:** Participants were asked to assess the generated videos based on four key criteria:
> 	- Reasonableness of the outpainted content.
> 	- Naturalness of the video.
> 	- Minimal distortion.
> 	- Clarity of the output.
> - **Scoring Rules:** Each participant assigned **a score of 1 to the best result and 0 to the others for each video**. The scores were then aggregated to derive the final comparisons.
> - **Scoring Results:**  The average score and standard deviation are reported for each method, as shown below.
>
> |       | M3DDM     | MOTIA     | HERO       |
> | ----- | --------- | --------- | ---------- |
> | Score | 2.17±2.55 | 3.17±1.40 | 34.75±1.91 |
> - **Analysis of User Study Results:**
> 	- **Method Perspective:** HERO achieved the highest user scores, with an average score of **34.75**, indicating that nearly 35 out of 40 videos were recognized as the best by users. This demonstrates HERO’s overwhelming advantage. The second and third rankings were MOTIA and M3DDM, respectively, aligning with quantitative metrics in Table 3.
> 	- **Video Perspective:** Analyzing individual video performance, 18 videos generated by HERO were unanimously rated as the best by all 12 participants. Additionally, HERO videos were deemed the best by at least 10 participants in **34 out of 40 videos**, highlighting its consistent performance.
> 	- **Participant Perspective:** Undergraduate and master’s degree participants showed a stronger preference for HERO compared to Ph.D. participants. While Ph.D. participants demonstrated a more critical perspective, they still selected HERO in over **75% of the videos** as the best-performing method.
> 	- **Observations:** During the evaluation, we observed significant shortcomings in M3DDM, where many outpainting results failed entirely, as acknowledged in Figure 9 and Section D of Appendix in  M3DDM paper. MOTIA and HERO almost never fail to produce outpainting results and the main difference between them lies in the quality of the output.
>
>
> - **Bad Case Study**:
> 	- For rapid human movements (e.g., quick head turns, fast leg lifts), HERO’s performance was suboptimal. On the _cat-girl.mp4_ video, HERO received 5 votes, MOTIA received 6 votes, and M3DDM received 1 vote. On the _kid-football.mp4_ video, HERO received 1 vote, MOTIA received 11 votes, and M3DDM received 0 votes. On the _pigs.mp4_ video, HERO, MOTIA, and M3DDM each received 4 votes. This is likely due to the challenge of handling fast-moving objects, which remains an area requiring further model optimization.
> 	- Additionally, based on our testing, HERO still struggles with certain artificial effects, such as the outpainting of animal limbs (e.g., multiple legs on an animal), the body of animals (e.g., camels or rhinoceroses), and complex indoor scenes, all of which require further refinement.
>
> We hope this additional user study adequately addresses your concerns and strengthens the reliability of our evaluations.

---

> ### Author Response · Authors · 2024-11-24
> **Response to Reviewer ddaE (Part3/3)**
>
> We have also included the score records for each method provided by the 12 participants.
>
> |     | M3DDM | MOTIA | HERO |
> | --- | ----- | ----- | ---- |
> | P1  | 0     | 3     | 37   |
> | P2  | 3     | 3     | 34   |
> | P3  | 0     | 4     | 36   |
> | P4  | 8     | 1     | 31   |
> | P5  | 3     | 2     | 35   |
> | P6  | 0     | 6     | 34   |
> | P7  | 0     | 3     | 37   |
> | P8  | 2     | 3     | 35   |
> | P9  | 1     | 5     | 34   |
> | P10 | 6     | 2     | 32   |
> | P11 | 1     | 2     | 37   |
> | P12 | 2     | 4     | 35   |
>
>
> **\[Q6. Artifacts in supplementary videos.\]**
>
>
> Thank you for your observation.
> - While we acknowledge the presence of some local jittering artifacts and object distortions, HERO demonstrates significant improvement over previous SoTA methods, as shown by both quantitative and qualitative results, as well as the user study.
> - We are actively working to refine the model and reduce these artifacts, focusing on advanced temporal modeling techniques and improving smooth transitions in dynamic scenes for future versions.
>
>
> **\[Q7. More evaluations on complicated scenes.\]**
>
> - We specifically evaluated the performance in extreme cases such as occlusions, object re-appearance and fast motions,  comparing HERO’s outpainted videos with real videos.
> - The results have been placed in the “Newly_Add” folder of the supplementary materials for your review.
>
>
> **\[Q8. Comments on Figure3.\]**
>
> We appreciate your attention to detail and thank you for carefully reviewing our manuscript.
> The figure labeling error in Figure 3 has been addressed as you suggested in the newly submitted manuscript.
>
>
>
>
> [1] Fan, F., Guo, C., Gong, L., Wang, B., Ge, T., Jiang, Y., ... & Zhan, J. (2023, October). Hierarchical masked 3d diffusion model for video outpainting. In _Proceedings of the 31st ACM International Conference on Multimedia_ (pp. 7890-7900).
>
> [2] Wang, F. Y., Wu, X., Huang, Z., Shi, X., Shen, D., Song, G., ... & Li, H. (2025). Be-your-outpainter: Mastering video outpainting through input-specific adaptation. In _European Conference on Computer Vision_ (pp. 153-168). Springer, Cham.
>
> [3] Guo, Y., Yang, C., Rao, A., Liang, Z., Wang, Y., Qiao, Y., ... & Dai, B. (2023). Animatediff: Animate your personalized text-to-image diffusion models without specific tuning. International Conference on Learning Representations (ICLR) 2024.
>
> [4] Zhang, L., Rao, A., & Agrawala, M. (2023). Adding conditional control to text-to-image diffusion models. In _Proceedings of the IEEE/CVF International Conference on Computer Vision_ (pp. 3836-3847).
>
> [5] Hu, L. (2024). Animate anyone: Consistent and controllable image-to-video synthesis for character animation. In _Proceedings of the IEEE/CVF Conference on Computer Vision and Pattern Recognition_ (pp. 8153-8163).
>
> [6] Yu, L., Cheng, Y., Sohn, K., Lezama, J., Zhang, H., Chang, H., ... & Jiang, L. (2023). Magvit: Masked generative video transformer. In _Proceedings of the IEEE/CVF Conference on Computer Vision and Pattern Recognition_ (pp. 10459-10469).

---

> > ### Comment · Reviewer_ddaE · 2024-12-03
> >
> > [Q1|Q3|Q4|Q6]: I am satisfied with the responses provided, and these weaknesses are resolved
> >
> > Q2: Video outpainting can have multiple plausible solutions with significant differences from each other. In such cases, do conventional perceptual metrics work effectively, or are there better alternatives to evaluate these kinds of outputs?
> >
> > Q5: While the user study results indicate a clear trend, the scale of the study remains limited. Additionally, the presentation of the results could be improved. It would be more appropriate to use a preferred rate metric rather than the current score-based metrics.
> >
> > Q7: Could you clarify what inputs were used for the newly added videos? Additionally, how do the baseline methods perform on these videos?
> >
> > Based on the other reviews and your current responses, I will raise my score to above 5 (I am still hesitating on the exact rating)

---

> ### Author Response · Authors · 2024-12-04
> **Response to Reviewer ddaE**
>
> **\[Q2.  Effectiveness of Conventional perceptual metrics.\]**
>
> Video outpainting allows less creative freedom compared to text-to-image and text-to-video generation tasks, making conventional perceptual metrics applicable under specific conditions.
>
> - **When conventional perceptual metrics work.** The effectiveness of conventional perceptual metrics **decreases as the outpainting ratio increases**. Based on this trend, our empirical findings indicate that when the outpainting ratio is below 2/3 (*e.g.*, video is extend by 1/3 on left and right side respectively based the middle content in the OPV setting) , these metrics remain generally reliable for quantitatively evaluating outpainting methods. In HERO, the outpainting ratio is within 2/3 in almost all quantitative experiments (except for the OPC setting).
> - **When conventional perceptual metrics don't work.** In cases of **ultrawide outpainting**, where the content is extended to multiple times the original width, these metrics become nearly ineffective. In such cases, we recommend employing human evaluation or specialized scoring models to **independently assess key aspects such as video coherence, naturalness, clarity, and distortion-free quality**, and then combining these scores for a more comprehensive evaluation.
>
>
>
> **\[Q5. Improvements on the user study.\]**
>
> - We appreciate that the ``preferred rate metric`` can more effectively highlight the performance differences between different methods and a larger scale of the user study could provide greater statistical significance.
> - However, due to the **practical challenges** of recruiting over 12 volunteers to individually rate 120 (40 × 3) videos for preference scores, we opted for the current score-based metrics given the scale of our user study. These metrics still reveal a clear trend.
> - Together with the **user study**, the **quantitative metrics** in Tables 1-3, the **qualitative comparison** in Figure 6, and the **videos** in the supplementary materials provide sufficient evidence to demonstrate the superiority of HERO.
>
>
>
> **\[Q7. The newly added videos.\]**
>
> - The videos in ``Newly_Add`` showcase the typical results of OPV (Vertical Outpainting). The input consists of the middle 50% of the content, while HERO extends both the left and right 25% borders.
> - For a clearer understanding, please refer to Figures 5 and 6, where the content inside the yellow dashed lines represents the input, and the areas outside are the outpainted regions.
> - In these videos, M3DDM fails in some outpainting cases, resulting in blurred borders, while MOTIA exhibits temporal instability and loss of detail. Some of these comparative results are similar to those shown in Figure 6.
>
>
>
> We hope the responses above help further address your concerns.
>
>
> HERO represents **the first effort to harness temporal modeling** for diffusion-based video outpainting, further strengthened by the valuable feedback from reviewers.
> We sincerely appreciate the reviewer’s participation and contributions.
>
> We hope the reviewer will thoughtfully consider HERO’s **technical contributions, robust experimental support, and its significant potential impact** on the future of video editing research.

---

### Official Review · Reviewer_C5r4 · 2024-11-04

**Soundness:** 3
**Presentation:** 3
**Contribution:** 3
**Rating:** 5
**Confidence:** 4

**Summary:**

This paper presents HERO (Harnessing the tEmpoRal modeling for diffusion-based Outpainting), a novel video outpainting method that addresses quality issues such as blurred details, local distortion, and temporal instability.HERO uses two key components: the temporal reference module ( TRM) and an interpolation-based motion modeling module (IMM).TRM provides reference features beyond the spatial dimension, while IMM stabilizes the generated frames. The authors have conducted extensive experiments on several benchmarks to demonstrate that HERO outperforms existing methods both qualitatively and quantitatively.

**Strengths:**

- Innovative Approach: HERO introduces a novel approach to video outpainting by focusing on temporal modeling, which is a significant advancement in the field.
- Temporal Reference Module (TRM): The TRM effectively provides reference features that enhance the spatial-temporal context, leading to improved outpainting quality.
- Interpolation-based Motion Modeling Module (IMM): The IMM stabilizes generated frames, reducing temporal instability and improving the overall quality of the outpainted video.
- Comprehensive Experiments: The paper includes extensive experiments on multiple benchmarks, providing strong empirical validation of the method's effectiveness.

**Weaknesses:**

- Limited Dataset Diversity: The experiments are primarily conducted on the DAVIS and YouTube-VOS datasets, which may not fully capture the diversity of real-world video content. I am curious about the model’s performance on ultra-high-resolution videos or other datasets with large and complex motion patterns. I would like to see more comprehensive experiments. Adding qualitative comparisons on real-world videos would be a big plus—for example, using the proposed algorithm to directly outpaint videos captured on a mobile phone. Specifically, can HERO perform well for 1080p videos, or higher resolution videos (2K or 4K), which are part of our daily lives? What is the runtime and computational complexity like on these scenarios? Also how well does the HERO perform for real-life scenarios in sports? For example, basketball and soccer in sports, or even badminton and table tennis in small resolution.

- Insufficient Error Analysis: The paper lacks a detailed error analysis that could provide insights into the method's limitations and failure cases. For example, will HERO have temporal discontinuous results? What are the reasons for these failure cases?

- Computational Efficiency: The computational efficiency of the method, especially in comparison to existing methods, is not thoroughly analyzed. I suggest that the authors provide a comparison of computational complexity, including metrics such as parameter count, FLOPs, runtime, and energy consumption.

- Hyperparameter Sensitivity: The sensitivity of the model to hyperparameters is not extensively discussed, which could impact its practical usability. For instance, is the diffusion model sensitive to hyperparameter choices?

- User Studies: I suggest that the authors include user studies. I would suggest that the authors could conduct a user study by finding a group of people (e.g., the size of 20 people) and counting the scores that these people give to the results obtained by the different methods.

- Comparison to Other Methods: The comparison to other methods is limited to a few benchmarks, and a more comprehensive comparison could strengthen the paper.

**Questions:**

Could you elaborate further on the significance of video outpainting. In my view, this area seems quite narrow, with arguably less value compared to video inpainting (minor point).

---

> ### Author Response · Authors · 2024-11-24
> **Response to Reviewer C5r4 (Part 1/4)**
>
> Thank you for your careful consideration of HERO and the thoughtful feedback you provided.
> Your insights has greatly contributed to enhancing the paper’s quality.
> We hope the clarifications below will address your concerns.
>
> **\[Q1. Limited Dataset Diversity.\]**
>
> - **Dataset Diversity and Generalization.** In our experiments, we extensively evaluated HERO not only on the DAVIS and YouTube-VOS datasets but also on the SSV2 dataset (Table 1), which contains 27k videos. Furthermore, Figure 5 presents a wide range of scenarios, including **landscapes, human figures (full-body, half-body, head-shot), swiftly moving vehicles, complex backgrounds, telefocus and near-focus videos, as well as cartoon videos**, demonstrating the diverse nature of the datasets used. These examples highlight that our method has been tested across a broad spectrum of content, showcasing its strong generalization capability and ensuring sufficient dataset diversity for real-world applicability.
> - **Real-life videos.** In the supplementary materials of our first submission, we already included 10 example videos, which cover real-life scenarios such as football and boxing. Additionally, we **further evaluated HERO's performance in extreme cases**, such as occlusions, object re-appearance, and fast motions. We compared HERO’s outpainted videos with real videos, and the results have **been placed in the "Newly_Add"** folder of the supplementary materials for your review.
> - **Regarding ultra-high-resolution videos**, due to our hardware memory constraints, we primarily focused on resolutions up to 512. HERO has been trained and evaluated on 128, 256, and 512 resolutions, and the consistent performance improvements across these scales suggest its potential for 2K or 4K videos, given suitable hardware.
> - **HERO has the potential for real-world applications**. For real-world applications, HERO can be further optimized for practical deployment by integrating quantization techniques and faster inference algorithms(e.g., LCM[7]), making it feasible for high-resolution and real-time scenarios like sports or mobile video outpainting.
>
>
>
> **\[Q2.  Computational Efficiency.\]**
>
>
> Thank you for raising the insightful point regarding model parameters and computational cost. We deeply appreciate it and have taken efforts to conduct thorough measurements.
>
> - In addition to the diffusion-based video outpainting baselines M3DDM[1] and MOTIA[2], we have also included two notable video generation methods, Animatediff[3] and AnimateAnyone[5], in our evaluation. These methods, whose network designs are widely referenced in the video generation field, share many similarities with M3DDM, MOTIA, and HERO.
> - We observed that many prior works[1,3,5] based on diffusion models **lack comprehensive statistics on parameters and computational requirements**. To address this gap, **we set up an environment to align benchmarks and fill this shortcoming in the community**. Furthermore, We have published our evaluation protocols, including the evaluation tools, video resolution details, hardware specifications, and metrics to establish a solid benchmark. **This paves a way for future research in the community to build upon**. For our evaluations,
> 	- **Evaluation Tool.** We used the ``calflops`` tool from GitHub to ensure standardized assessments.
> 	- **Video Resolution.** Video outpainting was conducted with a resolution of 256x256 to maintain fairness and consistency, and all other parameters adhered to the default settings of the open-source methods.
> 	- **Device.** Model speed tests were performed on an NVIDIA A100 GPU.
> 	- **Metrics.** We measured dimensions such as computational cost, parameter count, inference time, and peak memory usage.
> 	- **Formats.** We report the mean and variance of the inference times based on 10 runs following 3 warmup iterations.
> - **This table below will be included in the main text as a foundational resource, serving as a solid reference for future researchers.  This will encourage future papers on to honestly report their parameter counts and computational costs, fostering healthy development within both the research community and the industry.**
>
> The statistical results are shown in the table below.
>
> |               | TFLOPS↓ | Parameter Size(M)↓ | Inference Time↓ | Peak Memory↓ |
> | ------------- | ------- | ------------------ | --------------- | ------------ |
> | AnimateDiff   | 250.36  | 1687.6             | 9.90±0.02s      | 15.07G       |
> | AnimateAnyone | 255.06  | 1906.4             | 12.57±0.04s     | 12.9G        |
> | M3DDM         | 245.44  | 1109.57            | 21.27±0.09 s    | 13.96G       |
> | MOTIA         | 259.44  | 1692.52            | 10.53±0.01s     | 14.28G       |
> | HERO          | 259.99  | 2399.09            | 13.00±0.23 s    | 31.05G       |

---

> ### Author Response · Authors · 2024-11-24
> **Response to Reviewer C5r4 (Part2/4)**
>
> From the table above, it can be observed that:
> - **HERO significantly increases parameter counts.** As noted in the **Limitations section**  of this paper, HERO does have shortcomings in terms of parameter count and memory usage. We have **truthfully**  reported these limitations to ensure users are aware of these factors. As shown in the table, HERO requires more resources, with a 41.74% increase in parameters and approximately a two-fold increase in memory.
> - **The increase in computational cost is trivial.** However, despite the significant increase in parameters, these additional parameters are involved in the computation only once during video outpainting and **do not require iterative processing** during the denoising stage, thus having a **negligible impact** on computational cost. In terms of computational cost (TFLOPS), HERO only increases by **0.2%** compared to the previous state-of-the-art (SoTA), MOTIA.
> - **The inference time.** Consequently, the inference time of our method remains comparable to MOTIA, with only a modest 23% increase. For video generation tasks, 13s is still within a reasonable range.
> - **Performance trade-off**. Nevertheless, HERO demonstrates significant improvements in both quantitative and qualitative results, as well as in newly added user studies. The trade-off in performance for enhanced outcomes is justified.
> - **Mitigation measures.** In situations with limited memory resources, if you want to use HERO, you can extract the features of modules like OFE and 3D-RefNet offline and use them during the denoising process, which can save up to half of the memory.
> - It is important to specifically note that M3DDM employs a coarse-to-fine strategy, which improves frame quality but requires generating multiple video frames, resulting in higher inference time.
>
> We hope the responses can address your concerns.
>
>
>
> **\[Q3.  Hyperparameter Sensitivity.\]**
>
> - The core contribution of HERO lies in explicitly leveraging temporal modeling to enhance video outpainting performance.In the Temporal Reference Module, a key parameter is $\alpha$. To address hyperparameter sensitivity, we designed $\alpha$ to be learnable, thereby eliminating the need for manual tuning. The optimal values of  $\alpha$  at different layers are shown in Fig. 7.
>
> - For hyperparameters such as the learning rate during training and the number of diffusion steps $T$ during inference, we followed the configurations provided by established open-source implementations. This approach ensured efficiency and avoided the high computational cost of extensive hyperparameter tuning during training.
>
>
>
> **\[Q4.  User Studies and Error Analysis.\]**
> Thank you for highlighting the value of human evaluations, which we agree are essential for assessing the perceived quality of video outpainting.
> The human evaluations are conducted as follows.
> - **Participant Demographics:** The user study involved 12 participants, consisting of 1 females and 11 males. The group had diverse educational backgrounds, including 2 Ph.D. holders, 8 master’s degree holders, and 2 bachelor’s degree holders, with participants from at least two different cities, reflecting varied growth environments.
> - **Experimental Setup:** A total of 40 videos were randomly selected from the DAVIS dataset for evaluation. Each video was masked by removing 25% of the content from both the left and right sides, resulting in a total of 50% masked video content.
> - **Comparison Methods:** The performance of three methods was compared: M3DDM, MOTIA, and HERO.
> - **Blind Testing:** We ensured that the participants were unaware of which method each video came from, and the arrangement of the videos from the three methods was shuffled for each round of scoring.
> - **Evaluation Criteria:** Participants were asked to assess the generated videos based on four key criteria:
> 	- Reasonableness of the outpainted content.
> 	- Naturalness of the video.
> 	- Minimal distortion.
> 	- Clarity of the output.
> - **Scoring Rules:** Each participant assigned **a score of 1 to the best result and 0 to the others for each video**. The scores were then aggregated to derive the final comparisons.
> - **Scoring Results:**  The average score and standard deviation are reported for each method, as shown below.
>
> |       | M3DDM     | MOTIA     | HERO       |
> | ----- | --------- | --------- | ---------- |
> | Score | 2.17±2.55 | 3.17±1.40 | 34.75±1.91 |

---

> ### Author Response · Authors · 2024-11-24
> **Response to Reviewer C5r4 (Part3/4)**
>
> - **Analysis of User Study Results:**
> 	- **Method Perspective:** HERO achieved the highest user scores, with an average score of **34.75**, indicating that nearly 35 out of 40 videos were recognized as the best by users. This demonstrates HERO’s overwhelming advantage. The second and third rankings were MOTIA and M3DDM, respectively, aligning with quantitative metrics in Table 3.
> 	- **Video Perspective:** Analyzing individual video performance, 18 videos generated by HERO were unanimously rated as the best by all 12 participants. Additionally, HERO videos were deemed the best by at least 10 participants in **34 out of 40 videos**, highlighting its consistent performance.
> 	- **Participant Perspective:** Undergraduate and master’s degree participants showed a stronger preference for HERO compared to Ph.D. participants. While Ph.D. participants demonstrated a more critical perspective, they still selected HERO in over **75% of the videos** as the best-performing method.
> 	- **Observations:** During the evaluation, we observed significant shortcomings in M3DDM, where many outpainting results failed entirely, as acknowledged in Figure 9 and Section D of Appendix in  M3DDM paper. MOTIA and HERO almost never fail to produce outpainting results and the main difference between them lies in the quality of the output.
>
> - **Bad Case Study**:
> 	- For rapid human movements (e.g., quick head turns, fast leg lifts), HERO’s performance was suboptimal. On the _cat-girl.mp4_ video, HERO received 5 votes, MOTIA received 6 votes, and M3DDM received 1 vote. On the _kid-football.mp4_ video, HERO received 1 vote, MOTIA received 11 votes, and M3DDM received 0 votes. On the _pigs.mp4_ video, HERO, MOTIA, and M3DDM each received 4 votes. This is likely due to the challenge of handling fast-moving objects, which remains an area requiring further model optimization.
> 	- Additionally, based on our testing, HERO still struggles with certain artificial effects, such as the outpainting of animal limbs (e.g., multiple legs on an animal), the body of animals (e.g., camels or rhinoceroses), and complex indoor scenes, all of which require further refinement.
>
> We hope this additional user study adequately addresses your concerns and strengthens the reliability of our evaluations.
>
> We have also included the score records for each method provided by the 12 participants.
>
> |     | M3DDM | MOTIA | HERO |
> | --- | ----- | ----- | ---- |
> | P1  | 0     | 3     | 37   |
> | P2  | 3     | 3     | 34   |
> | P3  | 0     | 4     | 36   |
> | P4  | 8     | 1     | 31   |
> | P5  | 3     | 2     | 35   |
> | P6  | 0     | 6     | 34   |
> | P7  | 0     | 3     | 37   |
> | P8  | 2     | 3     | 35   |
> | P9  | 1     | 5     | 34   |
> | P10 | 6     | 2     | 32   |
> | P11 | 1     | 2     | 37   |
> | P12 | 2     | 4     | 35   |
>
>
>
> **\[Q6.  More Baselines.\]**
>
> - In recent years, the state-of-the-art (SoTA) methods for video outpainting have been MAGVIT, M3DDM, and MOTIA. This paper incorporates **all these SoTA methods as well as their baselines** for comprehensive evaluation.
> - To the best of our knowledge, HERO stands as the most comprehensively evaluated method in the field of video outpainting, providing the broadest experimental comparisons to date.
> - We further referred to the experimental results presented in the appendix of MAGVIT, incorporating its performance under various backbones and task settings. These results have been included into the following table, enriching the comparisons presented in Table 1 of this paper.
>
> | Method       | OPC↓     | OPV↓    | OPH↓    | AVG↓     |
> | ------------ | -------- | ------- | ------- | -------- |
> | MAGVIT-B-UNC | 67.5     | 27.3    | 36.2    | 43.66    |
> | MAGVIT-B-FP  | 213.2    | 81.2    | 86.3    | 126.90   |
> | MAGVIT-B-MT  | 38.8     | 23.3    | 26.1    | 29.39    |
> | MAGVIT-L-MT  | 21.1     | 16.8    | 17.0    | 18.3     |
> | M3DDM        | 19.2     | 14.5    | 14.3    | 16.0     |
> | HERO         | **18.9** | **9.4** | **9.1** | **12.4** |
>
>
> **\[Q7.  The Significance of Video Outpainting.\]**
>
> - Video outpainting addresses **real industrial needs**, which served as the motivation for our work.
> - With the rise of multi-device content consumption, videos often fail to fit diverse screen sizes and aspect ratios, resulting in suboptimal viewing experiences. Platforms like YouTube and TikTok commonly address this by enlarging videos, adding blurred backgrounds, and cropping to fit. While practical, this method leaves visible blurred edges, diminishing user immersion.
> - We are confident that **the reviewers have encountered numerous examples of such videos**, highlighting the substantial demand for improved solutions.
> - With HERO, we envision replacing these blurred-edge backgrounds with intelligently outpainted content, marking a significant technical leap forward.
> - In addition, video outpainting can also be applied in scenarios such as advertising, gaming, and VR/AR applications.

---

> ### Author Response · Authors · 2024-11-24
> **Response to Reviewer C5r4 (Part4/4)**
>
> We hope the clarifications can resolve your concerns.
>
> [1] Fan, F., Guo, C., Gong, L., Wang, B., Ge, T., Jiang, Y., ... & Zhan, J. (2023, October). Hierarchical masked 3d diffusion model for video outpainting. In _Proceedings of the 31st ACM International Conference on Multimedia_ (pp. 7890-7900).
>
> [2] Wang, F. Y., Wu, X., Huang, Z., Shi, X., Shen, D., Song, G., ... & Li, H. (2025). Be-your-outpainter: Mastering video outpainting through input-specific adaptation. In _European Conference on Computer Vision_ (pp. 153-168). Springer, Cham.
>
> [3] Guo, Y., Yang, C., Rao, A., Liang, Z., Wang, Y., Qiao, Y., ... & Dai, B. (2023). Animatediff: Animate your personalized text-to-image diffusion models without specific tuning. International Conference on Learning Representations (ICLR) 2024.
>
> [4] Zhang, L., Rao, A., & Agrawala, M. (2023). Adding conditional control to text-to-image diffusion models. In _Proceedings of the IEEE/CVF International Conference on Computer Vision_ (pp. 3836-3847).
>
> [5] Hu, L. (2024). Animate anyone: Consistent and controllable image-to-video synthesis for character animation. In _Proceedings of the IEEE/CVF Conference on Computer Vision and Pattern Recognition_ (pp. 8153-8163).
>
> [6] Yu, L., Cheng, Y., Sohn, K., Lezama, J., Zhang, H., Chang, H., ... & Jiang, L. (2023). Magvit: Masked generative video transformer. In _Proceedings of the IEEE/CVF Conference on Computer Vision and Pattern Recognition_ (pp. 10459-10469).
>
> [7] Luo, S., Tan, Y., Huang, L., Li, J., & Zhao, H. (2023). Latent consistency models: Synthesizing high-resolution images with few-step inference. _arXiv preprint arXiv:2310.04378_.

---

### Author Response · Authors · 2024-11-24
**General Response**

We appreciate all reviewers for their valuable comments and suggestions. Your insights have significantly contributed to improving the quality of the paper.
Building on this constructive feedback, we emphasize that:
-  HERO is the first work to hit the issue of insufficient temporal modeling in existing diffusion methods and explicitly harness temporal modeling to improve video outpainting performance.
- HERO reflects our innovation in extending the remarkable performance of diffusion models from image generation to the challenging domain video editing.


We have carefully addressed the comments and highlighted the key improvements as follows:
- We have compared our method with related approaches in terms of parameters and computational complexity, filling a gap in the community’s benchmarking. **This provides a foundation for future research**, with this comparison table serving as a key reference.
- We conducted a **human evaluation with 12 participants** to assess HERO and baseline methods. This makes HERO’s experimental results more robust.
- We have added the **mean and standard deviation** to the quantitative evaluation on the DAVIS dataset, enhancing the reliability of the results.
- We have added video demos of HERO in **extreme scenarios** (*e.g.*, occlusions, object re-appearance, and fast motions) to the supplementary materials (in the ‘Newly_Add’ folder) for your review,

For each of the reviewers’ specific comments, please refer to the corresponding responses.

Minor changes have already been directly incorporated into the revised manuscript, while the newly added tables and sections will be included in the appropriate sections of the final version.

---

> ### Author Response · Authors · 2024-12-01
> **Looking forward to your valuable response.**
>
> Dear Reviewers,
>
> We have carefully submitted our response and actively participated in the author-reviewer discussion phase.
> As the deadline approaches, most reviewers have not yet engaged in the discussion.
>
> We believe that further interaction could help clarify any issues and contribute to the improvement of HERO.
>
> We look forward to your valuable feedback.
>
> Best Regards,
>
> Authors of HERO

---

### Meta-Review · Area_Chair_4EkP · 2024-12-16

**Metareview:**

The paper was reviewed by four experts providing initially unanimous negative ratings (<6).

The authors provided responses and three experts kept their initial ratings while one improved to "6: marginally above".

Most of the reviewers and the AC agree that the paper is technically sound, but at the same time the contributions are rather limited and despite the extra information and new results provided by the authors three out of four reviewers remained on the negative with their ratings (<6).

Given the above, the AC finds the paper below the acceptance bar an invites the authors to benefit from the received feedback and further improve their work.

**Additional Comments On Reviewer Discussion:**

The authors provided responses and new results and while one reviewer was convinced to update the rating to "6", three other reviewers remained on the negative side.

---

### Decision · Program_Chairs · 2025-01-22

Reject